# DEGREE-CORRECTED RICCI CURVATURE FOR NETWORKS

## ABSTRACT

Networks are useful in understanding complex systems. Recently, discrete curvature measures, adapted from smooth manifolds to discrete landscape of networks, have been applied to network data analysis, such as community detection, with encouraging results. A fundamental hypothesis made for using these curvature measures is the diagonal-dominating principle: *the curvature measures are consistently larger within a community than between different communities*. However, this principle may not hold under all statistical network models. We investigate three existing network curvatures, which satisfy the diagonal-dominating principle under the stochastic block model (SBM) but not under its widely-used degree-corrected version, the degree-corrected block model (DCBM). Observing that these curvature measures are heavily influenced by degree heterogeneity, we propose a new curvature measure, Degree-Corrected Ricci Curvature (DCRC), specifically designed to account for degree heterogeneity. Theoretically, we prove that DCRC always satisfies the diagonal-dominating principle under both SBM and DCBM. We also provide large-deviation bounds and uncertainty quantification. Empirically, we use DCRC to preprocess a network by filtering out low-curvature edges; and we show that this preprocessing step can improve the performance of state-of-art community detection algorithms.

## 1 INTRODUCTION

Networks are ubiquitous across various fields, ranging from biological systems (Koutrouli et al., 2020) and social interactions (Ji et al., 2022) to technological infrastructures (Rosas-Casals et al., 2007) and cosmic formations (De Regt et al., 2018). These networks provide a robust framework for deciphering the complex relationships and dynamics inherent in diverse systems. Recent advancements in network analysis have highlighted the significant role of discrete curvature (Sia et al., 2019). Discrete curvature, adapting geometric concepts such as Ricci curvature (Ricci & Levi-Civita, 1900; Do Carmo & Flaherty Francis, 1992)—traditionally applied to smooth manifolds—to the discrete landscape of networks, has led to the development of various curvature measures, each offering unique insights into network properties.

Specific forms of discrete curvature for edges, such as the Ollivier-Ricci Curvature (ORC, Ollivier, 2007; 2009) and the Forman Ricci Curvature (FRC, Forman (2003)), have been pivotal in studying network transport efficiency and robustness. They provide a gauge of how networks deviate from being geometrically flat, which in turn offers insights into the overall connectivity and resilience of the network. The Balanced Forman Curvature (BFC, Topping et al., 2021), and Lower Ricci Curvature (LRC, Park & Li, 2025), refinements of the FRC, have been particularly effective in pinpointing bottleneck structures and critical connections within networks. The Jaccard curvature (Pal et al., 2017), a similarity metric between nodes, provides a low-complexity approximation of ORC. Theoretical studies of network curvatures, especially for ORCs, have primarily focused on connections to their continuous counterpart, the Ricci curvature of smooth manifolds (Lin & Yau, 2010; Lin et al., 2011; Bauer et al., 2013; Trillos & Weber, 2023). Their applications range from understanding internet topology (Ni et al., 2015) to differentiating cancer networks (Sandhu et al., 2015), addressing challenges in graph neural networks (Nguyen et al., 2023), and graph subsampling while preserving its property (Wu et al., 2023). There are also curvatures defined on nodes for planar graphs, such as Bakry-Emery curvature (Pouryahya et al., 2016), Gaussian curvature (Narayan & Saniee, 2011) and Gromov curvature (Higuchi, 2001).

Notably, these curvature-based methods have shown great potential in community detection (Sia et al., 2019; Park & Li, 2025), a critical area in network analysis. Community detection typically focuses on identifying subgroups within networks where nodes are more densely connected internally than with the rest of the network (Dey et al., 2022). The utility of curvature measures in this context lies in their ability to distinguish edges that span communities—typically exhibiting lower curvature—from those within communities, which tend to have higher curvature. This distinction has led to the development of methods that simplify community structures by iteratively removing edges with the most negative curvature or, alternatively (Sia et al., 2019; Fesser et al., 2023), by removing edges below a certain curvature threshold in a single step (Park & Li, 2025). These methods complement traditional community detection techniques (Jin, 2015; Chen et al., 2018; Zhang et al., 2020; Qin & Rohe, 2013) and offer a novel approach to refining community structures.

Unfortunately, the fundamental assumption, we term as *diagonal-dominating property*, behind these methods—*the curvature measures tend to be higher within a community than across*—may not always hold. We present a simulation example in Figure 1. The first five panels present histograms of several curvatures (the aforementioned BFC, FRC, Jaccard, LRC, as well as DCRC to be introduced) of a sample network generated from the Stochastic Block Model (SBM), where curvatures within community edges (blue) are indeed higher than those across communities (red).

SBM, while commonly used for its simplicity, does not capture the severe degree heterogeneity in real networks (i.e., even in the same community, some nodes have much higher degrees than others). Degree-Corrected Block Model (DCBM, Karrer & Newman, 2011) generalizes SBM by introducing node-wise degree parameters to capture the difference of popularity among nodes in the same community, and this model has been adopted by many recent works in community detection (Zhao et al., 2012; Jin, 2015; Qin & Rohe, 2013; Wang et al., 2020). The right five panels in Figure 1 reveal that the existing network curvature measures are inappropriate to use in the DCBM, because they are heavily influenced by degree heterogeneity.

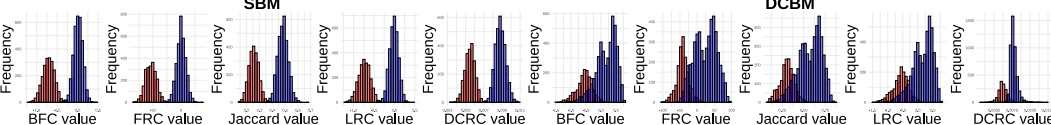

Figure 1: Curvature histograms for SBM (left five) and DCBM (right five), respectively. A desirable curvature should lead to well-separated histograms under both SBM and DCBM. Existing curvatures (BFC, FRC, Jaccard, LRC) do not satisfy this requirement for DCBM, but our proposed DCRC does.

To address these shortcomings, we propose a novel curvature measure, the Degree-Corrected Ricci Curvature (DCRC), specifically designed to handle degree heterogeneity effectively. DCRC not only addresses the limitations observed with traditional curvature measures but also demonstrates superior properties for community detection within DCBM frameworks. Our key contributions are:

**Theory.** To the best of our knowledge, this is the first theoretical study of network curvature under random graph models, such as SBM and DCBM. We analyze when the *diagonal-dominating property* holds, and provide the first theoretical guarantees for curvature-based community detection. Our results also include large deviation bounds and asymptotic distribution that enables uncertainty quantification for curvature-based downstream tasks.

**Methodology.** We introduce DCRC, a new curvature measure that removes the impact of degree heterogeneity via a simple normalization trick derived from theoretical analysis under DCBM. While motivated by DCBM, the method generalizes to broader network settings.

**Application.** We demonstrate that DCRC-based preprocessing improves existing community detection methods on both simulated and real networks.

Code and data availability, proofs, and additional experimental details are provided in the Appendix.

## 2 LIMITATIONS OF EXISTING NETWORK CURVATURES

**Existing network curvatures.** The idea of using curvature to study discrete objects, specifically networks, traces back to Forman (2003), which introduced the later-called FRC. Later, the ORC was

proposed (Ollivier, 2007), adapting Ricci curvature via optimal transpose. These notions have since inspired a wide range of studies in graph theory, discrete geometry and differential geometry.

The use of curvature for community detection emerged with Sia et al. (2019), who observed that in simple random graphs such as those generated by the SBM, within-community edges tend to have higher curvature than across-community edges. This property, which we refer to as the *diagonal-dominating property*, motivates pruning-based strategies: by removing edges with the most negative curvature values, the remaining network better reflects the underlying community structure. Subsequent works have introduced refinements of FRC, such as BFC and LRC, to improve scale-invariance and scalability. Among these, FRC and LRC are more tractable and computationally efficient; in contrast, BFC and ORC, while effective, are less amenable to theoretical analysis and more computationally demanding.

We now define the curvature notions rigorously and introduce the necessary notation. For clarity and simplicity in presentation, this paper focuses on an unweighted graph $G = (V, E)$, where $V$ is a set of nodes and $E$ is a set of edges, but the framework can be generalized to a weighted graph without significant complication. Let $(ij)$ be an edge connecting node $i$ and node $j$, we denote the degree of $i$, i.e., the number of edges of node $i$, by $n_i$, the number of shared neighbors of $i, j$, i.e., the number of triangles based on $(ij)$, by $n_{ij}$, and the size of union of neighbors of $i$ and $j$ by $u_{ij}$. Among the various curvature measures, we define FRC and LRC below due to their scalability and analytical tractability. We omit BFC and ORC, whose more intricate formulations make them less suitable for theoretical analysis in our context.

**Definition 1** (FRC, Jaccard, and LRC). *The FRC, Jaccard, and LRC of edge $(ij)$ are defined as*

$$\text{FRC}(ij) = 4 - n_i - n_j + 3n_{ij}, \quad \text{Jaccard}(ij) = \frac{n_{ij}}{u_{ij}},$$

$$\text{LRC}(ij) = \frac{2}{n_i} + \frac{2}{n_j} - 2 + 2\frac{n_{ij}}{\max(n_i, n_j)} + \frac{n_{ij}}{\min(n_i, n_j)}.$$

**Limitations of FRC, Jaccard, and LRC.** To understand the limitations of these curvatures, especially in the context of community detection, we investigate the *diagonal-dominating property*. If this property holds, then removing edges with low curvature values making the underlying community structure more distinct and easier to detect. As shown in the left half of Figure 1, this property holds under SBM, which assumes homogeneous node degrees and provides a useful baseline for analyzing community structure. However, this property may not hold in other settings–particularly when there is degree heterogeneity. To examine this scenario, we recall the definition of DCBM.

Let $A \in \mathbb{R}^{n \times n}$ be the adjacency matrix of an undirected network with $n$ nodes. Suppose that the nodes partition into $K$ non-overlapping communities $\mathcal{C}_1, \mathcal{C}_2, \ldots, \mathcal{C}_K$. For each node $i$, let $\pi_i \in \{0, 1\}^K$ be its membership vector, such that if $i \in \mathcal{C}_k$, then $\pi_i(k) = 1$ and $\pi_i(\ell) = 0$ for all $\ell \neq k$.

**Definition 2** (DCBM). *Let $\theta_i \in (0, 1)$ be the degree heterogeneity parameter of node $i$, and let $P \in \mathbb{R}^{K \times K}$ be a symmetric nonnegative matrix with unit diagonals and $P_{k\ell} \leq 1$ for $k \neq \ell$. The upper triangle of $A$ (excluding the diagonal) contains independent Bernoulli random variables, where*

$$\mathbb{P}(A(i, j) = 1) = \theta_i \theta_j \cdot \pi_i' P \pi_j, \qquad 1 \leq i < j \leq n. \tag{1}$$

When $\theta_i \equiv \theta$ are constant, DCBM degenerates to SBM as a special case.

**A motivating example.** Consider a DCBM network of 2 communities with 100 nodes in each, $P = [1, 0.5; 0.5, 1]$. Let $\theta_i = 1$ for all but the last node, where $\theta_{200} = \beta$ controls the degree of heterogeneity. To reduce the impact of randomness and simplify the experiment, we only consider the population version of all curvatures, that is, replace all random variables, i.e., $n_i, n_j, n_{ij}$ by their expectations. In this population version, we examine the difference between within-community curvature and across-community curvature. If the difference is positive, the *diagonal-dominating property holds*, which is desirable for curvature-based community detection. The detailed theoretical derivation is delayed to Section 3, and we present the results in Figure 2. All curvatures are rescaled to be comparable, without affecting the conclusion, as the main focus is whether the difference is positive or negative. We observe that when the network is homogeneous, i.e., $\beta = 1$, all curvatures admit the *diagonal-dominating property*. However, as $\beta$ increases, FRC, Jaccard, and LRC quickly lose this property, because they, by design, fail to account for heterogeneity. In contrast, our proposed curvature DCRC, to be discussed in next section, remains diagonal-dominating

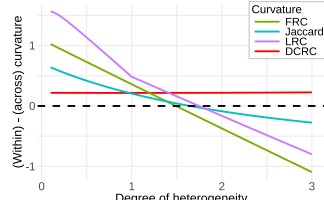

Figure 2: Within-community curvature minus across-community curvature for FRC, Jaccard, LRC, and DCRC (population version normalized by network size) versus degree heterogeneity $\beta$.

regardless of the degree heterogeneity. Next, we will formally introduce DCRC and investigate its theoretical properties in details.

## 3 DEGREE-CORRECTED RICCI CURVATURE (DCRC)

To address the limitations of existing curvatures, we propose DCRC, a new curvature.

**Definition 3** (DCRC). *The DCRC of edge $(ij)$ is defined as* $\mathrm{DCRC}(i,j) = \frac{n_{ij}}{n_i n_j}$.

To see the rationale, we recall the definition of DCBM and introduce a Bernoulli probability matrix $\Omega \in \mathbb{R}^{n \times n}$, where $\Omega_{ij} = \theta_i \theta_j \cdot \pi_i' P \pi_j$. The DCBM framework implies that $\mathbb{E}[A] = \Omega - \mathrm{diag}(\Omega)$. We regard $\Omega$ as the 'signal' matrix of $A$. When the networks are properly dense, $A - \Omega$ is negligible compared to $\Omega$ (this will be made rigorous in our theory). Hence, $n_i = \sum_k A_{ik} \approx \sum_j \Omega_{ik}$, and $n_{ij} = \sum_k A_{ik} A_{kj} \approx \sum_k \Omega_{ik} \Omega_{kj}$. We combine them with the definition of $\Omega$. It follows that

$$\mathrm{DCRC}(i,j) \approx \frac{\sum_k \Omega_{ik}\Omega_{kj}}{\left(\sum_k \Omega_{ik}\right)\left(\sum_k \Omega_{jk}\right)} = \frac{\cancel{\theta_i}\cancel{\theta_j}\sum_k (\pi_i' P \pi_k)(\pi_j' P \pi_k)}{\left(\cancel{\theta_i} \sum_k \theta_k \pi_i' P \pi_k\right)\left(\cancel{\theta_j} \sum_k \theta_k \pi_j' P \pi_k\right)}.$$

The effects of $\theta_i$ and $\theta_j$ have been canceled in the numerator and denominator. This explains how DCRC adjusts for degree heterogeneity. While degree parameters still exist in the denominator, they are in terms of an aggregated effect of all $\theta_k$'s, which is shared by all pairs of nodes $(i,j)$. As a result, when we apply DCRC, this aggregated effect of $\theta_k$'s has little impact on the relative order of curvatures within and between communities. The benefit of DCRC is further supported by the example in Figure 2. As $\beta$ increases, DCRC always maintains the diagonal-dominating property.

In the remaining of this section, we provide theoretical analysis of DCRC under the DCBM framework, including population analysis, large deviation bounds, and asymptotic distribution.

**3.1 The diagonal-dominating property of DCRC.** To establish this property, we introduce a population counterpart of DCRC. Write $\Theta = \mathrm{diag}(\theta_1, \theta_2, \ldots, \theta_n) \in \mathbb{R}^{n \times n}$ and $\Pi = [\pi_1, \pi_2, \ldots, \pi_n]' \in \mathbb{R}^{n \times K}$. We can re-write the DCBM model in a matrix form:

$$A = \Omega - \mathrm{diag}(\Omega) + W, \qquad \Omega = \Theta \Pi P \Pi' \Theta, \qquad W = A - \mathbb{E}[A]. \tag{2}$$

We call $\Omega$ the "signal" matrix, which carries all information of the community memberships. Introduce a vector $\eta := K\|\theta\|_1^{-1} \Pi' \Theta \mathbf{1}_n \in \mathbb{R}^K$ such that $\eta(k) = K(\sum_{i \in \mathcal{C}_k} \theta_i)/\|\theta\|_1$, and a diagonal matrix $G := \|\theta\|^{-2} \Pi' \Theta^2 \Pi \in \mathbb{R}^{K \times K}$ such that $G(k,k) = (\sum_{i \in \mathcal{C}_k} \theta_i^2)/\|\theta\|^2$, $1 \le k \le K$. Define

$$\mathrm{DCRC}^*(i,j) := \frac{n_{ij}^*}{n_i^* n_j^*}, \qquad \text{with} \quad \begin{cases} n_i^* = K^{-1}\theta_i\|\theta\|_1(\pi_i' P \eta), \\ n_{ij}^* = \theta_i \theta_j \|\theta\|^2 (\pi_i' P G P \pi_j) \end{cases} \tag{3}$$

In Lemma 1 below, we will show that $(n_i^*, n_j^*, n_{ij}^*)$ are the main-order terms in the expectation of $(n_i, n_j, n_{ij})$. Therefore, DCRC$^*$ serves as a population counterpart of DCRC. The following lemma provides a simple, explicit form of DCRC$^*$:

**Theorem 1** (Population DCRC). *For any $1 \le i \ne j \le n$,*

$$\mathrm{DCRC}^*(i,j) = \frac{K^2 \|\theta\|^2}{\|\theta\|_1^2} \cdot M_{k\ell}, \qquad \text{with } M = [\mathrm{diag}(P\eta)]^{-1} P G P [\mathrm{diag}(P\eta)]^{-1}. \tag{4}$$

This theorem confirms that the population DCRC is not affected by the individual degree parameters of node $i$ and node $j$. Furthermore, it implies: *the diagonal dominating property holds for the population DCRC, if the diagonal entries of $M$ are larger than the off-diagonal entries of $M$.*

The matrix $M$ can be computed easily for given DCBM parameters. The following corollary gives an interesting special case where the matrix $P$ has only two parameters $a$ and $b$ and the community sizes are balanced (the motivating example in Figure 2 belongs to this case with $K = 2$), a case commonly studied in the DCBM literature. Our results imply that the diagonal dominating property is always satisfied by the population DCRC in this case:

**Corollary 1** (Diagonal-dominating). *Under the DCBM model, suppose $\theta_i$'s are i.i.d. generated from a distribution whose support is at $[\alpha_n, h_1\alpha_n]$, for some $\alpha_n > 0$ and a constant $h_1 \geq 1$, and $\pi_i$'s are i.i.d. drawn from the uniform distribution over $\{e_1, e_2, \ldots, e_K\}$. As $n \to \infty$, $M$ converges to $M_0 := K \cdot [\mathrm{diag}(P\mathbf{1}_K)]^{-1}P^2[\mathrm{diag}(P\mathbf{1}_K)]^{-1}$ in probability. Furthermore, when the diagonal entries of $P$ are equal to $a$ and the off-diagonal entries are all equal to $b$, as long as $a \neq b$, the diagonal entries of $M_0$ are larger than the off-diagonal entries of $M_0$. As a result, $\mathrm{DCRC}^*(i, j)$ for two nodes in the same community must be larger than $\mathrm{DCRC}^*(i, j)$ for two nodes in distinct communities—the diagonal dominating property is satisfied.*

**3.2 Population analysis of DCRC.** We formally analyze the asymptotic properties of $\mathrm{DCRC}^*$. This requires a few technical conditions.

**Assumption 1.** *There exist constants $c_1, C_1, c_2, C_2, c_3, C_3 > 0$ such that $c_1 K^{-1}\|\theta\|_1 \leq \sum_{j \in \mathcal{C}_k} \theta_j \leq C_1 K^{-1}\|\theta\|_1$ and $c_2 K^{-1}\|\theta\|^2 \leq \sum_{j \in \mathcal{C}_k} \theta_j^2 \leq C_2 K^{-1}\|\theta\|^2$ for all $k \in [\![1, K]\!]$, and $c_3 > 0$ such that $c_3 < \lambda_{\min}(PGP) \leq \lambda_{\max}(PGP) < C_3$.*

**Assumption 2.** *There exists $C_4 \in (0, 1)$ such that $\theta_{\max} \leq C_4$. In addition, we have $K\theta_{\max}\|\theta\|_1^{-1} = o(1)$ and $K\theta_{\max}^2\|\theta\|^{-2} = o(1)$.*

**Assumption 3.** *There exists $c_5 > 0$ such that for all $k \in [\![1, K]\!]$, $|e_k'P\eta| > c_5$.*

**Assumption 4.** $\log(n)/[\theta_{\min}\|\theta\|_1] = o(1)$ *and* $\log(n)/[\theta_{\min}^2\|\theta\|^2] = o(1)$.

In Assumption 1, the first two inequalities imply that $K$ communities have balanced total degrees. In the special case where $\theta_i$'s are all in the same order, these inequalities are satisfied as long as the sizes of the $K$ communities are of the same order. The third inequality in Assumption 1 implies that $PGP$ is well conditioned. In a special case where $\theta_i$'s are i.i.d. drawn from a distribution, this reduces to requiring that the community matrix $P$ is well-conditioned, which is a commonly used assumption in the literature of SBM and DCBM (e.g., Jin, 2015). In Assumption 2, the requirement that $\theta_{\max} \leq C_4$ is mild. To understand the other two requirements, we consider a simple case where $K$ is finite, in which these requirements translate to $\theta_{\max} \ll \sum_{i=1}^n \theta_i$ and $\theta_{\max}^2 \ll \sum_{i=1}^n \theta_i^2$. This means that the degree parameter of any single node cannot be excessively dominant with respect to other nodes, which is a mild condition. Assumption 4 is about network sparsity. Suppose all $\theta_i$'s are at the order of $\alpha_n$. Then, this assumption reduces to $\alpha_n \gg n^{-1/4}[\log(n)]^{1/4}$. It implies that the average node degree has to grow with $n$ at a speed of at least $\sqrt{n\log(n)}$. As we will explain in equation 5 and the text therein, this condition is essential for any curvature metrics to be useful and cannot be further relaxed. Under these assumptions, we can calculate simplified expressions for the moments of $n_i$, $n_j$, and $n_{ij}$.

**Lemma 1.** *Under the DCBM model in equation 1-equation 2, suppose that Assumptions 1-2 hold. Consider two distinct nodes $i$ and $j$. The following statements are true:*

- $\mathbb{E}[n_i] = K^{-1}\theta_i\|\theta\|_1(\pi_i'P\eta) - \theta_i^2$, *and* $\mathrm{Var}(n_i) = O\big(K^{-1}\theta_i\|\theta\|_1\pi_i'P\eta\big)$.

- $\mathbb{E}[n_{ij}|A_{ij} = 1] = \theta_i\theta_j\|\theta\|^2(\pi_i'PGP\pi_j) - \theta_i\theta_j(\theta_i^2 + \theta_j^2)\pi_i'P\pi_j$.

- $\mathrm{Var}(n_{ij}|A_{ij} = 1) = O\big(\theta_i\theta_j\|\theta\|^2(\pi_i'PGP\pi_j)\big)$.

The following theorem examines the order the population DCRC. They will be useful for establishing large-deviation bounds and asymptotic distributions:

**Theorem 2** (Order of population DCRC). *Under the DCBM model, suppose that Assumptions 1-3 hold. When $i$ and $j$ are in the same community, $\frac{c_2 K\|\theta\|^2}{C_1^2\|\theta\|_1^2\|P\|_1^2} \leq \mathrm{DCRC}^*(i, j) \leq \frac{C_3 K^2\|\theta\|^2}{c_5^2\|\theta\|_1^2}$. When $i$ and $j$ are in distinct communities, $\mathrm{DCRC}^*(i, j) \leq \frac{c_2 K(P^2)_{k\kappa}\|\theta\|^2}{c_1^2\|\theta\|_1^2}$.*

**3.3 Large-deviation bounds for DCRC.** We characterize the deviation of DCRC from its population version. We first establish the following large deviation bounds for $n_i$, $n_j$, and $n_{ij}$.

**Lemma 2.** *Under the DCBM model, suppose that Assumptions 1-3 hold. Let $\delta > 0$. For $i, j$ such that $i \neq j$, with probability $1 - o(n^{-\delta})$, the following hold conditionally on $A_{ij} = 1$:*

- $\left| n_i - K^{-1}\theta_i\|\theta\|_1(\pi_i'P\eta) \right| \leq \frac{2\delta}{3}\log(n) + \left[ \frac{2\delta(\pi_i'P\eta)}{K}\theta_i\|\theta\|_1\log(n) \right]^{1/2}$.

- $\left| n_{ij} - \theta_i\theta_j\|\theta\|^2(\pi_i'PGP\pi_j) \right| \leq \frac{2\delta}{3}\log(n) + \|\theta\| \left[ 2\delta\theta_i\theta_j(\pi_i'PGP\pi_j)\log(n) \right]^{1/2}$.

Using the bounds in Lemma 2, we are equipped to derive concentration bounds for $\mathrm{DCRC}(i,j)$.

**Theorem 3.** *Under the DCBM model, suppose that Assumptions 1-4 hold. Let $\delta > 0$. For $i, j$ such that $i \neq j$, define $D_{ij} = |\mathrm{DCRC}(i,j) - \mathrm{DCRC}^*(i,j)|$. With probability $1 - o(n^{-\delta})$, the following inequalities hold conditionally on $A_{ij} = 1$:*

- $D_{ij} \leq \frac{cC_3K^2\|\theta\|^2}{c_5^2\|\theta\|_1^2} \left( \left[ \frac{\log(n)}{\theta_i\theta_j\|\theta\|^2} \right]^{\frac{1}{2}} + \left[ \frac{\log(n)}{\min\{\theta_i,\theta_j\}\|\theta\|_1} \right]^{\frac{1}{2}} \right)$, *if $i$ and $j$ are in the same community;*

- $D_{ij} \leq \frac{cc_2K(P^2)_{k\kappa}\|\theta\|^2}{c_1^2\|\theta\|_1^2} \left( \left[ \frac{\log(n)}{\theta_i\theta_j\|\theta\|^2} \right]^{\frac{1}{2}} + \left[ \frac{\log(n)}{\min\{\theta_i,\theta_j\}\|\theta\|_1} \right]^{\frac{1}{2}} \right)$, *if $i$ and $j$ belong to distinct communities $k$ and $\kappa$ (respectively).*

If we compare the order of the above bounds with the order of the population DCRC in Theorem 2, then we find that $\mathrm{DCRC}(i,j)$ will concentrate at $\mathrm{DCRC}^*(i,j)$ as long as the two terms inside the brackets are $o(1)$ as $n \to \infty$. This is equivalent to requiring

$$\theta_i\theta_j\|\theta\|^2 \gg \log(n), \qquad \min\{\theta_i,\theta_j\}\|\theta\|_1 \gg \log(n). \tag{5}$$

When $\theta_i$'s are all at the same order, equation 5 states that the average node degree should grow with $n$ at a speed faster than $\sqrt{n}$ —since this order can be $o(n)$, it permits moderately sparse networks.

**Remark**. This sparsity requirement is essential for the success of any curvature metrics. The effective sample size in all curvature metrics is determined by how many common neighbors two nodes share, whose expectation is at the order of $n\alpha_n^4$ if all $\theta_i$'s are at the order of $\alpha_n$. In comparison, the average node degree is at the order of $n\alpha_n^2$. Therefore, if we want $n\alpha_n^4$ to tend to infinity, we must need $n\alpha_n^2$ to be at least $\sqrt{n}$ .

**3.4 Asymptotic distribution of DCRC and uncertainty quantification.** Finally, we determine the asymptotic distribution of the DCRC in Theorem 4 and 5. We introduce the following assumptions on the network sparsity, which is akin to saying that $C_4 = o(1)$.

**Assumption 5.** $\theta_{\max} = o(1)$.

**Assumption 6.** *For all $1 \leq i \leq n$, $K^{-1}\|\theta\|_1\theta_i\pi_i'P\eta \gg 1$ and $\theta_i\theta_j\|\theta\|^2(\pi_i'PGP\pi_j) \gg 1$.*

To clarify these assumptions, we remark that in the context of SBM (with $\theta_i = \alpha_n$ for all $1 \leq i \leq n$), Assumption 6 implies that $n\alpha_n^4 \gg 1$, which means that the network's average degree $n\alpha_n^2$ must satisfy $n\alpha_n^2 \gg \sqrt{n}$. Therefore, we require that the network be sufficiently dense. This conditions comes from the use of $n_{ij}$, which counts the number of triangles passing by the edge $(i,j)$; ensuring that $n_{ij} \gg 1$ requires the network to be sufficiently dense.

**Theorem 4.** *Under the DCBM model in equation 1-equation 2, suppose that Assumptions 1-6 hold. Let $s_n^2 := \frac{\sum_{k\neq i,j} \Omega_{ki}\Omega_{kj}(1-\Omega_{ki})(1-\Omega_{kj})}{\left( \sum_{k=1}^n \Omega_{ki}\Omega_{kj} \right)^2}$. Then, conditionally on $A_{ij} = 1$, $T_{4,n} = \frac{\mathrm{DCRC}(i,j) - \mathrm{DCRC}^*(i,j)}{\mathrm{DCRC}^*(i,j) \cdot s_n} \xrightarrow{\mathcal{L}} \mathcal{N}(0,1)$.*

To apply Theorem 4 for uncertainty quantification, we also introduce an estimator of the asymptotic variance of $\mathrm{DCRC}(i,j)$ and establish the asymptotic normality of the self-normalized DCRC:

**Theorem 5.** *Under the DCBM model in equation 1-equation 2, suppose that Assumptions 1-6 hold. Let $\hat{v}_n^2 = \mathrm{DCRC}(i,j)^2/(A^2)_{ij}$. Then, conditionally on $A_{ij} = 1$, $T_{5,n} = \frac{\mathrm{DCRC}(i,j) - \mathrm{DCRC}^*(i,j)}{\hat{v}_n} \xrightarrow{\mathcal{L}} \mathcal{N}(0,1)$.*

From Theorem 5, we can derive a confidence interval for $\mathrm{DCRC}(i,j)$ and subsequently a threshold for curvature-based clustering. In Appendix D.1, we provide empirical support to Theorems 4 and 5.

## 4 SIMULATIONS

We conducted simulations to verify the strength of DCRC on assisting existing community detection algorithms. We simulated DCBM using parameter values specified in each experiment section. The notations for DCBM are the same as in the previous sections, with the exception of $\gamma$, which denotes the proportions of community sizes.

In the following experiments, each simulated graph is preprocessed by removing edges with curvature values smaller than a certain threshold. Then, on the preprocessed graph, we apply six existing representative community detection algorithms with diverse methodological approaches, which are used in comparative evaluation (Jin et al., 2021). These six methods are: Convexified Modularity Maximization (CMM, Chen et al., 2018), Latent Space Clustering via Distance (LSCD, Ma et al., 2020), Normalized Spectral Clustering (OCCAM, Zhang et al., 2020), Regularized Spectral Clustering (RSC, Qin & Rohe, 2013), Spectral Clustering on Ratios-of-Eigenvectors (SCORE, Jin, 2015), and its enhanced version, SCORE+ (Jin et al., 2021). The performance of community detection is evaluated by the clustering error rate (equation 25).

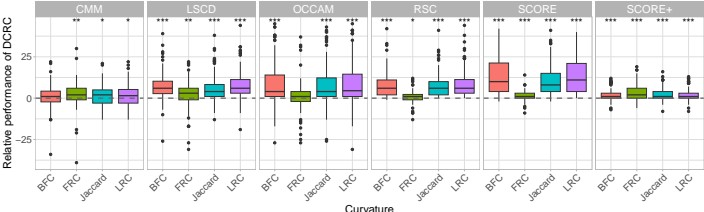
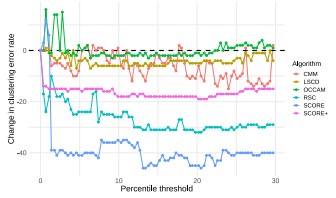

Figure 3: Relative performance of DCRC compared to BFC, FRC, Jaccard, and LRC, faceted by algorithms and curvatures.

Figure 4: Clustering error rate change across DCRC percentile threshold.

**Experiment 1 - Comparison of five curvatures:** In this experiment, we generated a hundred independent graphs with total node number $n = 150$, $k = 3$ communities of size proportion to $\gamma = (0.1, 0.2, 0.7)$, degree heterogeneity parameter $\theta \sim \text{Unif}(0.5, 1.5)$, and a probability matrix $P$ with diagonal entries equal to 0.6 and off-diagonal entries equal to 0.3. We first compared the performances of DCRC-based preprocessing method from other curvature-based preprocessing methods. We omitted ORC from the following analyses due to its high computational cost (cubic in number of nodes). After computing edge curvatures on the graph, we removed edges with DCRC values below a tuned threshold in percentile. Figure 3 shows the relative performance of DCRC compared with other curvature measures. Relative performance is defined as the difference between the improvement (of clustering error) over the baseline achieved by DCRC-based preprocessing and that achieved by an alternative curvature-based preprocessing. The x-axis lists the curvature measures being compared with DCRC, and values greater than zero indicate that DCRC provides greater improvement. Stars above each box plot indicate whether the mean relative performance is significantly greater than zero, based on a one-sample t-test, with 1-3 stars corresponding to p-values less than $0.05, 0.01, 0.001$. Overall, DCRC demonstrates significant improvement over most curvature-based preprocessing methods, with the exception of BFC under CMM and FRC under OCCAM. Even in these cases, however, the median remains above zero, indicating that DCRC outperforms the alternative curvature measure in more than half of the replicates.

**Experiment 2 - Robustness of DCRC preprocessing to thresholds:** Secondly, we evaluated the performance change of DCRC preprocessing across different percentile threshold. We conducted the same pipeline as Experiment 1. Figure 4 depicts the change in clustering error rate from that of the raw graph. Since we are using clustering error rate as performance metric, negative values in the plot indicate performance improvement over the baseline community detection algorithm. It appears that the results are initially unstable across all algorithms but begin to stabilize around 5% percentile. The performance of OCCAM shows the least improvement, while SCORE benefits the most from DCRC-based preprocessing. A key observation is that most algorithm exhibit either comparable or improved performance compared to the baseline after the 5% percentile threshold, demonstrating robustness of DCRC-based preprocessing to the choice of cutoff. We compared the performance of DCRC with other curvature measures across various threshold in Figure S1.

**Experiment 3 - Performance of DCRC across varying DCBM settings:** For the third experiment, we accessed the performance of DCRC preprocessing across different DCBM graph configurations. The parameters used for generating different DCBM graphs is provided in Table S1 and the distribution of DCRC for each graph configuration is given in Figure S2. For each graph configuration, we generated a hundred independent graphs and followed the same procedure as in Experiment 1. Figure 5 presents the improvement in community detection performance achieved by DCRC-based preprocessing relative to the baseline. While the magnitude of improvement varies across different DCBM graph conditions, DCRC consistently provides significant gains across all seven graph settings and six community detection algorithms considered in this experiment.

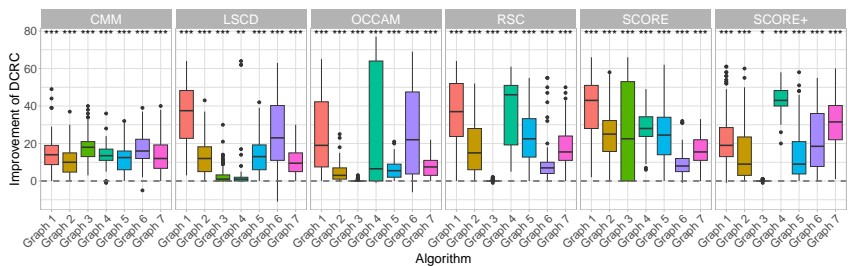

Figure 5: Improvement of DCRC over baseline, faceted by algorithms and graphs.

Appendix D.1 contains additional simulations for varying node size and number of communities.

## 5 APPLICATIONS

We assessed the practical utility of our DCRC through an application to real-world datasets: the Caltech, Simmons, and Political blog datasets. The Caltech and Simmons datasets represent Facebook social networks in September 2005 at California Institute of Technology and Simmons University, respectively (Red et al., 2011). The nodes indicate Facebook users at each university, and the edges represent "friend" relationship between two different users. For the following analysis, we used the community structure suggested by Red et al. (2011) as the ground truth community structure. As a different case, Political blog dataset is a single-day snapshot of citations among different political blogs in 2005 gathered by Adamic & Glance (2005). Each node represents a political blog and an edge denotes that at least one of the blogs cited the other. The community membership of each node is assigned from the blog's ideology classification.

For real-world dataset analysis, we use both the DCRC value and its estimated variance for DCRC preprocessing step since real data is noisier with edges that are false positive or false negative. Hence, we preprocessed the network data by deleting edges characterized by both low DCRC value and low variability, aiming to eliminate more reliable across-community edges. For other curvature measures, however, we only use the curvature value as the threshold, since the variance of these curvatures is not given in literature, further highlighting our theoretical contribution. Then, we use the same six community detection algorithms as before and report the clustering error rate.

**Facebook network at California Institute of Technology:** Caltech dataset has 590 nodes and 12,822 edges, with eight communities. Table 1 shows the result of applying different curvature preprocessing method to six community detection algorithms. Bold values show the best performance across different curvature metrics in a specific community detection algorithm, and the underlined value indicates the best performance across all curvature and all algorithms. Comparing the clustering error rate between curvatures, DCRC-based preprocessing achieves the lowest error rate compared to other curvatures. Although DCRC is not the best when applied with LSCD, it shows a significant performance improvement over the baseline. In fact, DCRC is the only curvature among the five that consistently improves all six algorithms.

**Facebook network at Simmons University:** Simmons dataset has 1,137 nodes and 24,257 edges with four communities. As shown in the second row of Table 1, although the FRC achieves the best overall performance, it does not consistently guarantee improvement over the baseline (see CMM).

Table 1: Clustering error rate comparison by curvature on three real networks

| Dataset | Curvature | CMM | LSCD | OCCAM | RSC | SCORE | SCORE+ |
|---------|-----------|-----|------|-------|-----|-------|--------|
| Caltech | Baseline | 125.0 | 100.0 | 195.0 | 213.0 | 179.0 | 100.0 |
| | BFC | 98.0 | 94.0 | 199.0 | 158.0 | 138.0 | 105.0 |
| | FRC | 87.0 | 103.0 | 157.0 | 170.0 | 148.0 | 102.0 |
| | Jaccard | 94.0 | 99.0 | 190.0 | 154.0 | 150.0 | 114.0 |
| | LRC | 97.0 | **93.0** | 197.0 | 155.0 | 137.0 | 104.0 |
| | DCRC | **82.0** | 97.0 | **118.0** | **131.0** | 134.0 | **94.0** |
| Simmons | Baseline | 139.0 | 135.0 | 267.0 | 359.0 | 268.0 | 127.0 |
| | BFC | 140.0 | 131.0 | 231.0 | 359.0 | 227.0 | 125.0 |
| | FRC | 139.0 | **127.0** | **217.0** | 311.0 | 218.0 | **120.0** |
| | Jaccard | 140.0 | 136.0 | 262.0 | 326.0 | 224.0 | 125.0 |
| | LRC | 137.0 | 132.0 | 238.0 | 355.0 | 229.0 | 125.0 |
| | DCRC | **131.0** | 131.0 | 222.0 | **281.0** | **213.0** | 123.0 |
| Polblog | Baseline | 61.0 | 60.0 | 59.0 | 394.0 | 58.0 | 51.0 |
| | BFC | 61.0 | 59.0 | 58.0 | 307.0 | 59.0 | 58.0 |
| | FRC | 62.0 | 58.0 | 59.0 | 377.0 | 59.0 | 52.0 |
| | Jaccard | 60.0 | 58.0 | 57.0 | 176.0 | 55.0 | 52.0 |
| | LRC | 61.0 | 58.0 | 59.0 | 351.0 | 63.0 | 57.0 |
| | DCRC | **57.0** | **51.0** | **54.0** | **55.0** | **52.0** | **50.0** |

However, DCRC consistently improves the performance over the baseline and achieves the best or second-best performance across all community detection algorithms.

**Political blog citation network:** Political blog dataset contains 12,222 nodes and 16,714 edges with two communities (liberal vs. conservative). As reported in the third row of Table 1, DCRC exceeds the other curvatures in all settings. For certain methods such as CMM, OCCAM, and SCORE+, curvature-based preprocessing generally yields marginal benefit. This pattern is probably due to the dataset's relatively simple community structure. Importantly, DCRC-based preprocessing consistently benefits all six community detection algorithms, significantly improves RSC, and achieves the best overall results when combined with SCORE+.

## 6 DISCUSSION

This study discusses how existing network curvature measures fail to account for degree heterogeneity. This is important, as it is natural for nodes to exhibit heterogeneous behavior in many real-world networks. Our main contribution is the introduction and the theoretical validation of DCRC, a novel network curvature that effectively addresses degree heterogeneity. Supported by rigorous theoretical guarantees, we proposed DCRC-based preprocessing that improves community detection performance. This is further evidenced by empirical results from both simulations and real-world datasets. The simulation experiments highlight DCRC's superior performance on DCBM graphs compared to previous curvature measures and demonstrate its effectiveness across six different community detection algorithms under a range of DCBM setting. The real-world analyses further affirm these findings, as DCRC-based preprocessing consistently achieves top-tier performance across most community detection algorithms on three real-world network dataset.

Several limitations of this study should be noted, since it provides promising future research directions. First, DCRC is defined only on an undirected, unweighted graph. As can be observed from gene networks to bank transaction networks, many real-world networks involve either directed or weighted edges. Hence, extending the definition of DCRC to directed and weighted graphs could significantly broaden its applicability to various real-world networks.Second, our analysis focuses on networks with discrete (non-overlapping) community memberships, while many real-world network exhibit overlapping structure. For example, genes often function in several biological pathways, and individuals typically belong to multiple social groups. Extending theoretical or empirical investigations of DCRC and other curvatures on networks with overlapping memberships will provide us with a method to reveal hidden structure that is not detected under hard clustering and allow for more informative analysis on real-world problems. Third, we mainly focus on community detection of network data in our study. By extending the study of various curvatures on different network analysis tasks, such as link prediction and network comparison, future research could give more comprehensive understanding of graph curvature measures.

## LLM USAGE DISCLOSURE.

The authors acknowledge the use of large language models (LLMs) to aid and polish writing. The authors have checked the LLM outputs and take full responsibility for the content.

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

## A  DATA AND CODE AVAILABILITY

Code is available at: `https://anonymous.4open.science/r/dcrc-68C4`

## B  PROOFS

Throughout the proofs, for a given node $i \in [\![1, n]\!]$ in the following proofs, we interchangeably use the notation $\pi_i$ and $e_{z(i)}$ to denote the (degenerate) membership vector of node $i$, where $z(i) \in [\![1, K]\!]$ maps $i$ to its corresponding network community.

### B.1  PROOF OF LEMMA 1

**Mean of $n_i$.**  We have, for all $i \in [\![1, n]\!]$:

$$\mathbb{E}[n_i] = \mathbb{E}\left[\sum_{j \neq i} A_{ij}\right] = \sum_{j=1}^{n} \Omega_{ij} - \Omega_{ii} = e_i'\Omega\mathbf{1}_n - e_i'\Omega e_i.$$

Therefore

$$\mathbb{E}[n_i] = e_i'\Omega\mathbf{1}_n - \theta_i^2 = \theta_i\pi_i P\Pi'\Theta\mathbf{1}_n - \theta_i^2 = K^{-1}\theta_i\|\theta\|_1(e_{z(i)}'P\eta) - \theta_i^2.$$

Furthermore, from Assumption 1, we see that:

$$e_i'\Omega\mathbf{1}_n = \theta_i \sum_{k,\ell} \sum_{j=1}^{n} \theta_j\pi_i(k)\pi_j(\ell)P_{k\ell} \geq cK^{-1}\theta_i\|\theta\|_1.$$

Therefore, using Assumption 2, we have

$$\frac{\theta_i^2}{e_i'\Omega\mathbf{1}_n} \leq \frac{K\theta_{\max}}{\|\theta\|_1} = o(1).$$

It follows that

$$\mathbb{E}[n_i] \asymp K^{-1}\theta_i\|\theta\|_1(e_{z(i)}'P\eta).$$

**Variance of $n_i$.**  We also have, for all $i \in [\![1, n]\!]$:

$$\mathrm{Var}(n_i) = \mathrm{Var}\left(\sum_{j \neq i} A_{ij}\right) = \sum_{j \neq i} \mathrm{Var}(A_{ij}) = \sum_{j \neq i} \Omega_{ij}(1 - \Omega_{ij}) \leq \sum_{j=1}^{n} \Omega_{ij} = e_i'\Omega\mathbf{1}_n.$$

Therefore,

$$\mathrm{Var}(n_i) \leq K^{-1}\theta_i\|\theta\|_1(e_{z(i)}'P\eta).$$

**Mean of $n_{ij}$.**  We have, for all $i, j \in [\![1, n]\!]$ with $i \neq j$:

$$\mathbb{E}[n_{ij}|A_{ij} = 1] = \mathbb{E}\left[\sum_{k \neq i,j}^{n} A_{ik}A_{jk}\right] = \sum_{k \neq i,j} \Omega_{ik}\Omega_{jk} = (\Omega^2)_{ij} - \Omega_{ij}(\Omega_{ii} + \Omega_{jj}).$$

Note that:

$$(\Omega^2)_{ij} = e_i'\Omega^2 e_j = e_i'\Theta\Pi P\Pi'\Theta^2\Pi P\Pi'\Theta e_j = \|\theta\|^2\theta_i\theta_j\pi_i'PGP\pi_j.$$

Hence,

$$\mathbb{E}[n_{ij}|A_{ij} = 1] = \theta_i\theta_j\|\theta\|^2(e_{z(i)}'PGPe_{z(j)}) - \theta_i\theta_j(\theta_i^2 + \theta_j^2)P_{z(i)z(j)}.$$

Furthermore, from Assumptions 1 and 2, we see that:

$$\frac{\theta_i\theta_j(\theta_i^2 + \theta_j^2)P_{z(i)z(j)}}{\theta_i\theta_j\|\theta\|^2(e_{z(i)}'PGPe_{z(j)})} \leq \frac{2\theta_{\max}^2}{\|\theta\|^2\lambda_{\min}(PGP)} = o(1).$$

It follows that

$$\mathbb{E}[n_{ij}|A_{ij} = 1] \asymp \theta_i\theta_j\|\theta\|^2(e_{z(i)}'PGPe_{z(j)}).$$

**Variance of $n_{ij}$.** We have, for all $i, j \in [\![1, n]\!]$ with $i \neq j$:

$$\text{Var}(n_{ij}|A_{ij} = 1) = \text{Var}\left(\sum_{k \neq i,j}^{n} A_{ik}A_{jk}\right) = \sum_{k \neq i,j} \text{Var}(A_{ik}A_{jk})$$

$$= \sum_{k \neq i,j} \Omega_{ik}\Omega_{jk}(1 - \Omega_{ik}\Omega_{jk}) \leq \sum_{k \neq i,j} \Omega_{ik}\Omega_{jk}.$$

Therefore, it follows that

$$\text{Var}(n_{ij}|A_{ij} = 1) \leq c\theta_i\theta_j\|\theta\|^2(e'_{z(i)}PGPe_{z(j)}).$$

$\square$

### B.2 PROOF OF THEOREM 1

In equation 3, we have defined $DCRC^*(i, j) := n^*_{ij}/(n^*_i n^*_j)$, where $(n^*_i, n^*_j, n^*_{ij})$ are the respective major terms in $(\mathbb{E}n_i, \mathbb{E}n_j, \mathbb{E}n_{ij})$. By Lemma 1, $n^*_i = K^{-1}\theta_i\|\theta\|_1 \cdot \pi'_i P\eta$ and $n^*_{ij} = \theta_i\theta_j\|\theta\|^2 \cdot \pi'_i PGP\pi_j$. We assume that $z(i) = k$ and $z(j) = \ell$ without loss of generality. It follows that

$$DCRC^*(i, j) = \frac{K^2\|\theta\|^2}{\|\theta\|_1^2} \cdot \frac{\pi'_i PGP\pi_j}{(\pi'_i P\eta)(\pi'_j P\eta)} = \frac{K^2\|\theta\|^2}{\|\theta\|_1^2} \cdot \frac{e'_k PGPe_\ell}{(e'_k P\eta)(e'_\ell P\eta)}$$

$$= \frac{K^2\|\theta\|^2}{\|\theta\|_1^2} \cdot \left[\text{diag}(P\eta)^{-1}\right]_{kk} (PGP)_{k\ell} \left[\text{diag}(P\eta)^{-1}\right]_{\ell\ell}$$

$$= \frac{K^2\|\theta\|^2}{\|\theta\|_1^2} \cdot \left[\text{diag}(P\eta)^{-1} \cdot (PGP) \cdot \text{diag}(P\eta)^{-1}\right]_{k\ell} = \frac{K^2\|\theta\|^2}{\|\theta\|_1^2} \cdot M_{k\ell}.$$

This proves the claim. $\square$

### B.3 PROOF OF LEMMA 1

In the definition of $M$, only $G$ and $\eta$ are random quantities under the assumptions of this lemma. Therefore, we first study $G$ and $\eta$.

Consider $G$. By definition, $G$ is a diagonal matrix, with

$$G(k, k) = \frac{\sum_{i \in \mathcal{C}_k} \theta_i^2}{\|\theta\|^2} = \frac{n^{-1}\sum_{i=1}^{n}(\alpha_n^{-2}\theta_i^2) \cdot 1\{z(i) = k\}}{n^{-1}\sum_{i=1}^{n}(\alpha_n^{-2}\theta_i^2)} := \frac{Y_k}{X_k}.$$

We recall that $(\alpha_n^{-2})\theta_i^2$ are i.i.d. variables whose support is in $[1, h_1^2]$. Let $\omega_0 = \mathbb{E}[\alpha_n^{-2}\theta_i^2]$. Since $\text{Var}(\alpha_n^{-2}\theta_i^2) < \infty$, by law of large numbers,

$$X_k \to \omega_0, \qquad \text{in probability.}$$

Moreover, $1\{z(i) = k\}$ are i.i.d. Bernoulli variables with a mean of $1/K$, and $z(i)$ is independent of $\theta_i$. It follows that $(\alpha_n^{-2}\theta_i^2) \cdot 1\{z(i) = k\}$ are i.i.d. variables whose mean is $\omega_0/K$ and whose variance is finite. Using law of large numbers again,

$$Y_k \to \omega_0/K, \qquad \text{in probability.}$$

We conclude that $G(k, k) \to K$ in probability, for each $1 \leq k \leq K$.

Consider $\eta$. By definition,

$$\eta(k) = \frac{K\sum_{i \in \mathcal{C}_k} \theta_i}{\|\theta\|_1} = \frac{K \cdot n^{-1}\sum_{i=1}^{n}(\alpha_n^{-1}\theta_i) \cdot 1\{z(i) = k\}}{n^{-1}\sum_{i=1}^{n}(\alpha_n^{-1}\theta_i)} := \frac{Z_k}{W_k}.$$

Let $\omega_1 = \mathbb{E}[\alpha_n^{-1}\theta_i]$. We can similarly show that

$$W_k \to \omega_1, \quad \text{in proability,} \qquad \text{and} \qquad Z_k \to K \cdot (\omega_1/K), \quad \text{in probability.}$$

We conclude that $\eta(k) \to 1$ in probability, for each $1 \leq k \leq K$.

We now show the claims. Our analysis of $G$ and $\eta$ has suggested that $G \to K \cdot I_K$ and $\eta \to \mathbf{1}_K$ in probability. It follows that

$$M = [\text{diag}(P\eta)]^{-1} PGP [\text{diag}(P\eta)]^{-1} \quad \to \quad M_0 := [\text{diag}(P\mathbf{1}_K)]^{-1} (K \cdot P^2)[\text{diag}(P\mathbf{1}_K)]^{-1}.$$

This proves the first claim about $M$.

When the diagonals of $P$ and $a$ and the off-diagonals are $b$, we can write $P = b\mathbf{1}_K\mathbf{1}_K' + (a-b)I_K$. It follows that

$$P\mathbf{1}_K = b\mathbf{1}_K(\mathbf{1}_K'\mathbf{1}_K) + (a-b)\mathbf{1}_K = [a + (K-1)b] \cdot \mathbf{1}_K,$$
$$P^2 = b\mathbf{1}_K\mathbf{1}_K'P + (a-b)P = b \cdot [2a + (K-2)b] \cdot \mathbf{1}_K\mathbf{1}_K' + (a-b)^2 I_K.$$

We plug them into the definition of $M_0$. It is easy to see that for all $k \neq \ell$,

$$M(k,k) = \frac{b \cdot [2a + (K-2)b] + (a-b)^2}{[a + (K-1)b]^2}, \qquad M(k,\ell) = \frac{b \cdot [2a + (K-2)b]}{[a + (K-1)b]^2}.$$

The diagonal entries of $M_0$ are strictly larger than the off-diagonal entries by a constant $(a-b)^2$. Since $\|M - M_0\| = o_{\mathbb{P}}(1)$, we conclude that with probability tending to 1, the diagonal entries of $M$ are strictly larger than the off-diagonal entries. $\qquad\square$

### B.4 PROOF OF THEOREM 2

We first obtain bounds for $e'_{z(i)}P\eta$ and $e'_{z(j)}PGPe_{z(j)}$ depending on the community memberships of $i$ and $j$.

When $i$ and $j$ belong to the same community $k$, we have:

$$e'_{z(i)}P\eta = e'_{z(j)}P\eta = \frac{K}{\|\theta\|_1} \sum_{\ell=1}^{K} \left( \sum_{u \in \mathcal{C}_\ell} \theta_u \right) P_{k\ell} \leq C_1 \|P\|_1,$$

$$e'_{z(j)}PGPe_{z(j)} = \frac{1}{\|\theta\|^2} \sum_{\ell=1}^{K} \left( \sum_{u \in \mathcal{C}_\ell} \theta_u^2 \right) P_{k\ell}^2 \geq \frac{c_2}{K}.$$

When $i$ and $j$ belong to distinct communities $k$ and $\kappa$ (respectively, with $k \neq \kappa$), we have:

$$e'_{z(i)}P\eta = \frac{K}{\|\theta\|_1} \sum_{\ell=1}^{K} \left( \sum_{u \in \mathcal{C}_\ell} \theta_u \right) P_{k\ell} \geq c_1,$$

$$e'_{z(j)}PGPe_{z(j)} = \frac{1}{\|\theta\|^2} \sum_{\ell=1}^{K} \left( \sum_{u \in \mathcal{C}_\ell} \theta_u^2 \right) P_{k\ell}P_{\kappa\ell} \leq C_2 (P^2)_{k\kappa}$$

As a result, when $i, j \in \mathcal{C}_k$, we can lower bound the population Degree-Corrected Ricci Curvature as follows:

$$DCRC^*(i,j) = \frac{K^2\|\theta\|^2(e'_{z(i)}PGPe_{z(j)})}{\|\theta\|_1^2(e'_{z(i)}P\eta)(e'_{z(j)}P\eta)} \geq \frac{c_2 K\|\theta\|^2}{C_1^2\|\theta\|_1^2\|P\|_1^2}.$$

We can also upper bound it, using Assumption 3:

$$DCRC^*(i,j) = \frac{K^2\|\theta\|^2(e'_{z(i)}PGPe_{z(j)})}{\|\theta\|_1^2(e'_{z(i)}P\eta)(e'_{z(j)}P\eta)} \leq \frac{C_3 K^2\|\theta\|^2}{c_5^2\|\theta\|_1^2}.$$

Conversely, when $i \in \mathcal{C}_k$ and $j \in \mathcal{C}_\kappa$ with $k \neq \kappa$, we upper bound the population Degree-Corrected Ricci Curvature as follows:

$$DCRC^*(i,j) = \frac{K^2\|\theta\|^2(e'_{z(i)}PGPe_{z(j)})}{\|\theta\|_1^2(e'_{z(i)}P\eta)(e'_{z(j)}P\eta)} \leq \frac{c_2 K(P^2)_{k\kappa}\|\theta\|^2}{c_1^2\|\theta\|_1^2}.$$

$\qquad\square$

## B.5 PROOF OF LEMMA 2

**Concentration of $n_i$.** Noting that $|A_{ij} - \Omega_{ij}| \leq 1$, we can apply Bernstein's inequality to obtain that for all $i \in [\![1, n]\!]$, with probability $1 - o(n^{-\delta})$:

$$|n_i - \mathbb{E}[n_i]| \leq \frac{2\delta}{3} \log(n) + \left[ \frac{2\delta(e'_{z(i)}P\eta)}{K} \theta_i \|\theta\|_1 \log(n) \right]^{1/2}.$$

Therefore, since $|n_i - e'_i \Omega \mathbf{1}_n| \leq |n_i - \mathbb{E}[n_i]| + e'_i \Omega e_i$ with $e'_i \Omega e_i = \theta_i^2$, we have with probability $1 - o(n^{-\delta})$:

$$\left| n_i - K^{-1}\theta_i \|\theta\|_1 (e'_{z(i)}P\eta) \right| \leq \frac{2\delta}{3} \log(n) + \left[ \frac{2\delta(e'_{z(i)}P\eta)}{K} \theta_i \|\theta\|_1 \log(n) \right]^{1/2}.$$

**Concentration of $n_{ij}$.** Noting that $|A_{ik}A_{jk} - \Omega_{ik}\Omega_{jk}| \leq 1$, we can apply Bernstein's inequality to obtain that for all $i, j \in [\![1, n]\!]$, with probability $1 - o(n^{-\delta})$:

$$|n_{ij} - \mathbb{E}[n_{ij}]| \leq \frac{2\delta}{3} \log(n) + \|\theta\| \left[ 2\delta\theta_i\theta_j(e'_{z(i)}PGPe_{z(j)}) \log(n) \right]^{1/2}.$$

Note that

$$\left| n_{ij} - (\Omega^2)_{ij}) \right| \leq |n_{ij} - \mathbb{E}[n_{ij}]| + \Omega_{ij}(\Omega_{ii} + \Omega_{jj})$$
$$\leq |n_{ij} - \mathbb{E}[n_{ij}]| + \theta_i\theta_j(\theta_i^2 + \theta_j^2).$$

Therefore, we have with probability $1 - o(n^{-\delta})$:

$$\left| n_{ij} - \theta_i\theta_j \|\theta\|^2 (e'_{z(i)}PGPe_{z(j)}) \right| \leq \frac{2\delta}{3} \log(n) + \|\theta\| \left[ 2\delta\theta_i\theta_j(e'_{z(i)}PGPe_{z(j)}) \log(n) \right]^{1/2}.$$

$\square$

## B.6 PROOF OF THEOREM 3

To simplify notations, we denote:

$$\bar{n}_i = K^{-1}\theta_i \|\theta\|_1 (e'_{z(i)}P\eta),$$
$$\bar{n}_{ij} = \theta_i\theta_j \|\theta\|^2 (e'_{z(i)}PGPe_{z(j)}),$$

and

$$\Delta_i = (n_i - \bar{n}_i)/\bar{n}_i,$$
$$\Delta_{ij} = (n_{ij} - \bar{n}_{ij})/\bar{n}_{ij}.$$

We can therefore write:

$$\left| \frac{n_{ij}}{n_i n_j} - \frac{\bar{n}_{ij}}{\bar{n}_i \bar{n}_j} \right| \leq \frac{\bar{n}_{ij}}{\bar{n}_i \bar{n}_j} \cdot \frac{\Delta_{ij} + \Delta_i + (1 + \Delta_j)\Delta_j}{(1 + \Delta_i)(1 + \Delta_j)}.$$

Based on Theorem 3, we know that with probability $1 - o(n^{-\delta})$,

$$\Delta_i \leq \frac{2K\delta \log(n)}{3\theta_i \|\theta\|_1 (e'_{z(i)}P\eta)} + \left[ \frac{2K\delta \log(n)}{\theta_i \|\theta\|_1 (e'_{z(i)}P\eta)} \right]^{1/2},$$

$$\Delta_{ij} \leq \frac{2\delta \log(n)}{3\theta_i\theta_j \|\theta\|^2 (e'_{z(i)}PGPe_{z(j)})} + \left[ \frac{2\delta \log(n)}{\theta_i\theta_j \|\theta\|^2 (e'_{z(i)}PGPe_{z(j)})} \right]^{1/2}.$$

Using Assumptions 1, 3, and 4, we obtain that

$$\Delta_i \leq \frac{C \log(n)}{\theta_i \|\theta\|_1} + \left[ \frac{C \log(n)}{\theta_i \|\theta\|_1} \right]^{1/2} \leq \left[ \frac{C \log(n)}{\theta_i \|\theta\|_1} \right]^{1/2} = o(1),$$

$$\Delta_{ij} \leq \frac{C \log(n)}{\theta_i\theta_j \|\theta\|^2} + \left[ \frac{C \log(n)}{\theta_i\theta_j \|\theta\|^2} \right]^{1/2} \leq \left[ \frac{C \log(n)}{\theta_i\theta_j \|\theta\|^2} \right]^{1/2} = o(1).$$

Notice that $DCRC^*(i,j) = \bar{n}_{ij}/(\bar{n}_i \bar{n}_j)$. It follows that:

$$\left| \frac{n_{ij}}{n_i n_j} - DCRC^*(i,j) \right| \le c \cdot DCRC^*(i,j) \cdot \left( \left[ \frac{\log(n)}{\theta_i \theta_j \|\theta\|^2} \right]^{1/2} + \left[ \frac{\log(n)}{\min\{\theta_i, \theta_j\} \|\theta\|_1} \right]^{1/2} \right).$$

It follows from Theorem 2 that if $i$ and $j$ belong to the same community:

$$\left| \frac{n_{ij}}{n_i n_j} - DCRC^*(i,j) \right| \le \frac{cC_3 K^2 \|\theta\|^2}{c_5^2 \|\theta\|_1^2} \cdot \left( \left[ \frac{\log(n)}{\theta_i \theta_j \|\theta\|^2} \right]^{1/2} + \left[ \frac{\log(n)}{\min\{\theta_i, \theta_j\} \|\theta\|_1} \right]^{1/2} \right),$$

and if $i$ and $j$ belong to distinct communities $k$ and $\kappa$ (respectively):

$$\left| \frac{n_{ij}}{n_i n_j} - DCRC^*(i,j) \right| \le \frac{cc_2 K(P^2)_{k\kappa} \|\theta\|^2}{c_1^2 \|\theta\|_1^2} \cdot \left( \left[ \frac{\log(n)}{\theta_i \theta_j \|\theta\|^2} \right]^{1/2} + \left[ \frac{\log(n)}{\min\{\theta_i, \theta_j\} \|\theta\|_1} \right]^{1/2} \right).$$

$\square$

### B.7  PROOF OF THEOREM 4

Fix $i$ and $j$. We introduce the following random variables:

$$U_i = \frac{n_i}{n_i^*}, \qquad U_j = \frac{n_j}{n_j^*}, \qquad V_{ij} = \frac{n_{ij} - n_{ij}^*}{n_i^* n_j^*}. \tag{6}$$

Recall that $DCRC(i,j) = n_{ij}/(n_i n_j)$ and $DCRC^*(i,j) = n_{ij}^*/(n_i^* n_j^*)$. It follows that

$$DCRC(i,j) = \frac{n_{ij}}{n_i^* n_j^*} \frac{1}{U_i U_j} = \left[ DCRC^*(i,j) + V_{ij} \right] \cdot \frac{1}{U_i U_j},$$

and

$$DCRC(i,j) - DCRC^*(i,j) = \frac{DCRC^*(i,j) \cdot (1 - U_i U_j)}{U_i U_j} + \frac{V_{ij}}{U_i U_j}.$$

Therefore, we have the following expression:

$$\frac{DCRC(i,j) - DCRC^*(i,j)}{DCRC^*(i,j) \cdot s_n} = \frac{1 - U_i U_j}{U_i U_j \cdot s_n} + \frac{V_{ij}}{DCRC^*(i,j) \cdot s_n} \times \frac{1}{U_i U_j}$$
$$\equiv I_1 + I_2 \times I_3.$$

To show that the left hand side converges to $\mathcal{N}(0,1)$, we only need to show the following results:

$$I_1 \xrightarrow{\mathbb{P}} 0, \qquad I_2 \xrightarrow{\mathcal{L}} \mathcal{N}(0,1), \qquad I_3 \xrightarrow{\mathbb{P}} 1. \tag{7}$$

Given equation 7, the claim of this theorem follows from elementary probability.

We now show equation 7. The first and third claims are both about $U_i U_j$, so we show them together. From Lemma 1 and the proof of Theorem 2, $n_i^* \asymp \theta_i \|\theta\|_1$, $\mathbb{E}[n_i] = n_i^* - \theta_i^2$, and $\mathrm{Var}(n_i) = O(\theta_i \|\theta\|_1)$. As a result,

$$\mathbb{E}\left[ (n_i - n_i^*)^2 \right] = \mathrm{Var}(n_i) + \left( \mathbb{E}[n_i] - n_i^* \right)^2 = O(\theta_i \|\theta\|_1) + O(\theta_i^4) = O(\theta_i \|\theta\|_1).$$

It follows that

$$\mathbb{E}[(U_i - 1)^2] = \frac{\mathbb{E}[(n_i - n_i^*)^2]}{(n_i^*)^2} = O\left( \frac{1}{\theta_i \|\theta\|_1} \right) = o(1), \tag{8}$$

where in the last equality we have used $\theta_i \|\theta\|_1 \to \infty$ (which is guaranteed by our assumption). The above implies that $U_i \xrightarrow{\mathbb{P}} 1$. Similarly, we can show that $U_j \xrightarrow{\mathbb{P}} 1$. It follows that

$$I_3 = \frac{1}{U_i U_j} \xrightarrow{\mathbb{P}} 1. \tag{9}$$

This proves the third claim in equation 7. To show the first claim in equation 7, we recall the definition of $s_n$ and comparing it with $n_{ij}^*$. It follows that

$$s_n^2 = \frac{1}{(n_{ij}^*)^2} \sum_{k \neq i,j} \Omega_{ki}\Omega_{kj}(1 - \Omega_{ki})(1 - \Omega_{kj})$$

$$= \frac{1}{(n_{ij}^*)^2} \sum_{k \neq i,j} \left[ \Omega_{ki}\Omega_{kj} - \Omega_{ki}^2\Omega_{kj} - \Omega_{ki}\Omega_{kj}^2 + \Omega_{ki}^2\Omega_{kj}^2 \right]$$

$$= \frac{1}{(n_{ij}^*)^2} \left\{ n_{ij}^* - \sum_{k \neq i,j} \left[ \Omega_{ki}^2\Omega_{kj} + \Omega_{ki}\Omega_{kj}^2 - \Omega_{ki}^2\Omega_{kj}^2 \right] \right\}. \tag{10}$$

Note that $\Omega_{ij} \leq \theta_i\theta_j$ for all $i, j$. Additionally, by Lemma 1 and Theorem 2, $n_{ij}^* \asymp \theta_i\theta_j\|\theta\|^2$. We combine these observations to obtain:

$$s_n^2 \geq \frac{n_{ij}^* - C\sum_k \theta_k^3\theta_i\theta_j}{(n_{ij}^*)^2} \geq \frac{n_{ij}^* - C\theta_i\theta_j\|\theta\|_3^3}{(n_{ij}^*)^2} \geq \frac{n_{ij}^* \cdot [1 - o(1)]}{(n_{ij}^*)^2} \geq \frac{C^{-1}}{\theta_i\theta_j\|\theta\|^2}, \tag{11}$$

where in the third inequality in equation 11 we have used $\|\theta\|_3^3 \leq \theta_{\max}\|\theta\|^2 = o(1) \cdot \|\theta\|^2$. We combine equation 11 with equation 8. It follows that

$$s_n^{-2}\, \mathbb{E}[(U_i - 1)^2] = O\left( \frac{\theta_i\theta_j\|\theta\|^2}{\theta_i\|\theta\|_1} \right) = O(\theta_{\max}^2) = o(1).$$

Similarly, we can show that $s_n^{-2}\, \mathbb{E}[(U_j - 1)^2] = o(1)$. We immediately conclude that

$$\frac{U_i - 1}{s_n} \xrightarrow{\mathbb{P}} 0, \qquad \text{and} \qquad \frac{U_j - 1}{s_n} \xrightarrow{\mathbb{P}} 0. \tag{12}$$

Notice that

$$I_1 = \frac{1 - U_iU_j}{U_iU_j \cdot s_n} = \frac{1}{U_iU_j} \times \frac{1 - U_i}{s_n} + \frac{1}{U_i} \times \frac{1 - U_j}{s_n}.$$

Here, $U_i$ and $U_j$ converge to 1 in probability, and $(1 - U_i)/s_n$ and $(1 - U_j)/s_n$ converge to 0 in probability. It follows that

$$I_1 \xrightarrow{\mathbb{P}} 0. \tag{13}$$

This proves the first claim in equation 7.

It remains to show the second claim in equation 7. Note that:

$$I_2 = \frac{V_{ij}}{\text{DCRC}^*(i, j) \cdot s_n} = \frac{(n_{ij} - n_{ij}^*)/(n_i^*n_j^*)}{n_{ij}^*/(n_i^*n_j^*) \cdot s_n} = \frac{n_{ij} - n_{ij}^*}{n_{ij}^* \cdot s_n}.$$

We use the expression of $s_n^2$ in equation 10. It yields that

$$I_2 = \frac{n_{ij} - n_{ij}^*}{\sqrt{\sum_{k \neq i,j} \Omega_{ki}\Omega_{kj}(1 - \Omega_{ki})(1 - \Omega_{kj})}}.$$

Meanwhile, we can decompose $n_{ij}$ as follows:

$$n_{ij} = \sum_{k=1}^{n} A_{ik}A_{kj} = \sum_{k \neq i,j} (\Omega_{ik} + W_{ik})(\Omega_{kj} + W_{kj})$$

$$= \sum_{k \neq i,j} \Omega_{ik}\Omega_{kj} + \sum_{k \neq i,j} \Omega_{ik}W_{kj} + \sum_{k \neq i,j} W_{ik}\Omega_{kj} + \sum_{k \neq i,j} W_{ik}W_{kj}$$

$$= \mathbb{E}[n_{ij}] + J_1 + J_2 + J_3.$$

Combining the above results gives

$$I_2 = \frac{(\mathbb{E}[n_{ij}] - n_{ij}^*) + J_1 + J_2 + J_3}{\sqrt{\sum_{k \neq i,j} \Omega_{ki}\Omega_{kj}(1 - \Omega_{ki})(1 - \Omega_{kj})}}. \tag{14}$$

We note that $J_3$ is a sum of independent, mean-zero random variables; furthermore, $\mathrm{Var}(W_{ik}W_{kj}) = \Omega_{ik}\Omega_{kj}(1-\Omega_{ik})(1-\Omega_{kj})$. We immediately have

$$\mathrm{Var}(J_3) = \sum_{k \neq i,j} \Omega_{ki}\Omega_{kj}(1-\Omega_{ki})(1-\Omega_{kj}).$$

Using this and equation 10, we have $s_n^2 = \mathrm{Var}(J_3)/(n_{ij}^*)^2$; and in equation 11, we have seen that $s_n^2 \geq [1-o(1)]/n_{ij}^*$. Combining these statements gives $\mathrm{Var}(J_3) \geq n_{ij}^* \cdot [1-o(1)]$. Meanwhile, it is easy to see that $\mathrm{Var}(J_3) \leq \sum_k \Omega_{ki}\Omega_{kj} \leq n_{ij}^*$. Hence, we re-write equation 14 as

$$I_2 = \frac{(\mathbb{E}[n_{ij}] - n_{ij}^*) + J_1 + J_2 + J_3}{\sqrt{\mathrm{Var}(J_3)}}, \qquad \text{where} \quad \mathrm{Var}(J_3) \sim n_{ij}^* \asymp \theta_i\theta_j\|\theta\|^2. \tag{15}$$

By Lemma 1, $\mathbb{E}[n_{ij}] - n_{ij}^* \leq C\theta_i\theta_j(\theta_i^2 + \theta_j^2)$. It follows that

$$\frac{\mathbb{E}[n_{ij}] - n_{ij}^*}{\sqrt{\mathrm{Var}(J_3)}} \leq \frac{C\theta_i\theta_j(\theta_i^2 + \theta_j^2)}{\sqrt{\theta_i\theta_j\|\theta\|^2}} \leq \frac{C\theta_{\max}^3}{\|\theta\|} = o(1). \tag{16}$$

Moreover,

$$\mathrm{Var}(J_1) = \sum_{k \neq i,j} \Omega_{ik}^2 \Omega_{kj}(1-\Omega_{kj}) \leq C \sum_k (\theta_i\theta_k)^2 \theta_k\theta_j \leq C\theta_i\theta_j\|\theta\|_3^3 = o(1) \cdot \mathrm{Var}(J_3).$$

Similarly, we can show that $\mathrm{Var}(J_2) = o(1) \cdot \mathrm{Var}(J_3)$. Combining these results gives

$$\frac{J_1 + J_2}{\sqrt{\mathrm{Var}(J_3)}} \xrightarrow{\mathbb{P}} 0. \tag{17}$$

We plug equation 16-equation 17 into equation 15. It is seen that

$$I_2 = o_{\mathbb{P}}(1) + \frac{J_3}{\sqrt{\mathrm{Var}(J_3)}}. \tag{18}$$

We now study $J_3$. It is the sum of $(n-2)$ independent but not identically distributed random variables. We apply Lyapunov's central limit theorem with $\delta = 2$ (it requires evaluating the sum of $(2+\delta)$th moments of these variables). By direct calculations,

$$\sum_{k \neq i,j} \mathbb{E}[W_{ik}^4 W_{kj}^4] \leq C \sum_k \theta_i\theta_k\theta_k\theta_j \leq C\theta_i\theta_j\|\theta\|^2.$$

As a result,

$$\frac{\sum_{k \neq i,j} \mathbb{E}[W_{ik}^4 W_{kj}^4]}{[\mathrm{Var}(J_3)]^2} \leq C \frac{\theta_i\theta_j\|\theta\|^2}{(\theta_i\theta_j\|\theta\|^2)^2} = o(1),$$

where the last equality is because our assumptions imply that $\theta_{\min}^2\|\theta\|^2 \to \infty$. This has verified the Lyapunov's condition. We immediately conclude that

$$\frac{J_3}{\sqrt{\mathrm{Var}(J_3)}} \xrightarrow{\mathcal{L}} \mathcal{N}(0,1). \tag{19}$$

The second claim in equation 7 follows from combining equation 19 with equation 18. This completes the proofs. $\square$

### B.8 PROOF OF THEOREM 5

By Theorem 4,

$$\frac{DCRC(i,j) - DCRC^*(i,j)}{DCRC^*(i,j)s_n} \xrightarrow{\mathcal{L}} \mathcal{N}(0,1), \tag{20}$$

where the variance can be estimated using:

$$\hat{v}_n^2 = DCRC(i,j)^2 \cdot \frac{1}{(A^2)_{ij}}.$$

Let's now show that $\hat{v}_n$ is a consistent estimator of $DCRC^*(i,j)s_n$. We have:

$$\frac{\hat{v}_n}{DCRC^*(i,j)s_n} = \frac{DCRC(i,j)}{DCRC^*(i,j)} \cdot \frac{1}{(A^2)_{ij}^{1/2}s_n} = \frac{DCRC(i,j)}{DCRC^*(i,j)} \cdot \frac{1}{(A^2)_{ij}^{1/2}\widetilde{s}_n} \cdot \frac{\widetilde{s}_n}{s_n}, \tag{21}$$

where we have introduced the quantity $\widetilde{s}_n := \left[(\Omega^2)_{ij} - \Omega_{ij}(\Omega_{ii} + \Omega_{jj})\right]^{-1/2}$. We will now show that each factor in the above expression converges to 1 as $n \to \infty$.

First, recognize that

$$s_n = \frac{1}{n_{ij}^*}\left(\sum_{k \neq i,j}\Omega_{ki}\Omega_{kj}(1 - \Omega_{ki})(1 - \Omega_{kj})\right)^{1/2}$$

$$\leq \frac{1}{n_{ij}^*}\cdot\left(\sum_{k \neq i,j}\Omega_{ki}\Omega_{kj}\right)^{1/2} \leq \frac{1}{(n_{ij}^*)^{1/2}} = \frac{1}{\theta_i\theta_j\|\theta\|^2(\pi_i'PGP\pi_j)}$$

$$\leq \frac{1}{c_5\theta_{\min}^2\|\theta\|^2} = o(1),$$

using Assumption 4. Therefore, combining the fact that $s_n = o(1)$ with Equation 20, we obtain from Slutsky's lemma that:

$$\frac{DCRC(i,j)}{DCRC^*(i,j)} \xrightarrow[n\to\infty]{\mathbb{P}} 1. \tag{22}$$

Second, we examine the factor $(A^2)_{ij}^{1/2}\widetilde{s}_n$. For convenience, we denote $\hat{s}_n := \left[(A^2)_{ij}\right]^{-1/2}$. Note that

$$\mathbb{E}\left[\left(\frac{(A^2)_{ij}}{(\Omega^2)_{ij} - \Omega_{ij}(\Omega_{ii} + \Omega_{jj})} - 1\right)^2\right] = \frac{\mathbb{E}\left[\left(\sum_{k \neq i,j}A_{ik}A_{jk} - \Omega_{ik}\Omega_{jk}\right)^2\right]}{\left(\sum_{k \neq i,j}\Omega_{ik}\Omega_{jk}\right)^2} = \frac{\mathrm{Var}\left(\sum_{k \neq i,j}A_{ik}A_{jk}\right)}{\left(\sum_{k \neq i,j}\Omega_{ik}\Omega_{jk}\right)^2}$$

$$= \frac{\sum_{k \neq i,j}\mathrm{Var}(A_{ik}A_{jk})}{\left(\sum_{k \neq i,j}\Omega_{ik}\Omega_{jk}\right)^2} = \frac{\sum_{k \neq i,j}\Omega_{ik}\Omega_{jk}(1 - \Omega_{ik}\Omega_{jk})}{\left(\sum_{k \neq i,j}\Omega_{ik}\Omega_{jk}\right)^2}$$

$$\leq \frac{1}{\sum_{k \neq i,j}\Omega_{ik}\Omega_{jk}} = \frac{1}{n_{ij}^* - \Omega_{ij}(\Omega_{ii} + \Omega_{jj})} = o(1).$$

Therefore, we obtain from Markov's inequality that $(A^2)_{ij}/\left[(\Omega^2)_{ij} - \Omega_{ij}(\Omega_{ii} + \Omega_{jj})\right] \xrightarrow{\mathbb{P}} 1$, which implies that $\widetilde{s}_n^2/\hat{s}_n^2 \xrightarrow{\mathbb{P}} 1$, and therefore that

$$\frac{1}{(A^2)_{ij}^{1/2}\widetilde{s}_n} = \frac{\hat{s}_n}{\widetilde{s}_n} \xrightarrow{\mathbb{P}} 1. \tag{23}$$

Third, we examine the factor $\widetilde{s}_n/s_n$. Note that

$$\frac{s_n^2}{\widetilde{s}_n^2} = \frac{1}{(n_{ij}^*)^2}\cdot\left(\sum_{k \neq i,j}\Omega_{ik}\Omega_{kj}(1 - \Omega_{ik}\Omega_{jk})\right)\cdot\left(\sum_{k \neq i,j}\Omega_{ik}\Omega_{jk}\right)$$

$$= \left[1 - \frac{\Omega_{ij}(\Omega_{ii} + \Omega_{jj})}{n_{ij}^*} - \frac{\sum_{k \neq i,j}\Omega_{ik}^2\Omega_{jk}^2}{n_{ij}^*}\right]\cdot\left[1 - \frac{\Omega_{ij}(\Omega_{ii} + \Omega_{jj})}{n_{ij}^*}\right]$$

$$= \left[1 - o(1) - \frac{\sum_{k \neq i,j}\Omega_{ik}^2\Omega_{jk}^2}{n_{ij}^*}\right]\cdot[1 - o(1)].$$

Here we will use the assumption that for all $u \in [\![1, n]\!]$, $\theta_u = o(1)$. This implies that $\Omega_{ik} = o(1)$, and therefore:

$$\frac{s_n^2}{\widetilde{s}_n^2} = 1 - o(1) \xrightarrow[n\to\infty]{} 1. \tag{24}$$

We can now combine equations 22-24 and equation 21 to obtain that $\hat{v}_n$ is a consistent estimator of $DCRC^*(i,j)s_n$:

$$\frac{\hat{v}_n}{DCRC^*(i,j)s_n} \xrightarrow[n\to\infty]{\mathbb{P}} 1.$$

Then, an additional application of Slutsky's lemma yields:

$$\frac{DCRC(i,j) - DCRC^*(i,j)}{DCRC(i,j) \cdot (A^2)_{ij}^{-1/2}} \xrightarrow{\mathcal{L}} \mathcal{N}(0,1).$$

$\square$

## C   DCRC-BASED PREPROCESSING ALGORITHM

In addition to introducing the novel DCRC measure, we propose a DCRC-based graph preprocessing method, adapted from the approach in Park & Li (2025), described in Algorithm S1. Here, $f$ denotes a function applied to the DCRC value. In the simulation studies, we used the identity function, meaning edges were removed only if their DCRC value was below a specified threshold. For the real data analysis, the function incorporated both the DCRC value and its variance estimate.

---
**Algorithm S1:** DCRC-based preprocessing method for community detection

---
**Input:** Raw network data: $G = (V, E)$
**Output:** Preprocessed network data $G' = (V, E')$
1  Calculate the DCRC for all edges;
2  Remove all edges based on specified criteria ($\alpha$) based on DCRC:
   $E' := \{(i,j) \in E : f(DCRC(i,j)) \geq \alpha\}$

---

## D   ADDITIONAL EXPERIMENTS AND DETAILS

The clustering error rate employed in this paper is as follows, widely used in multiple literature (Jin, 2015; Gao et al., 2018; Jin et al., 2021).

$$\text{Clustering Error Rate} = \min_{\tau:\text{permutation over } \{1,2,\dots,K\}} \frac{1}{n} \sum_{i=1}^{n} \mathbf{1}\{\tau(\ell_i) \neq \ell_i\}. \tag{25}$$

All experiments presented in the paper were conducted using a computing cluster, utilizing nodes equipped with AMD EPYC 9354 32-Core Processors and up to 200 GB per node. The total computation time is estimated as follows: approximately 1 hour for Experiments 1 and 2, about 1 hour and 40 minutes for Experiment 3, and roughly 46 hours for the real data analysis.

### D.1   SIMULATION

The graphs used in these additional results are the same as those specified in the main paper.

**Additional Results for Experiment 1 & 2:** Figure S1 illustrates the performance of various curvature metrics on different community detection algorithms across a range of percentile thresholds. From the figure, we observe that certain curvature metrics outperform DCRC at specific thresholds - for example, FRC in CMM and BFC in OCCAM. However, DCRC consistently achieves the lowest clustering error rate across a wide range of thresholds, whereas the performance of other curvature metrics tend to fluctuate more significantly. These results suggest that DCRC offers more stable and reliable improvements to community detection performance compared to other curvature-based preprocessing methods.

**Additional Results for Experiment 3:** Table S1 summarizes the parameters of the seven DCBM graphs used in the main text. In the *Parameters* column, (*x, y*) indicate the range of the Uniform distribution from which each $\theta_i$ is randomly sampled. *A* specifies the diagonal entry of the $P$ matrix, while *B* denotes the off-diagonal entry. Each graph has total $n = 150$ nodes with $k = 3$ communities where community size ratio is 1:2:7.

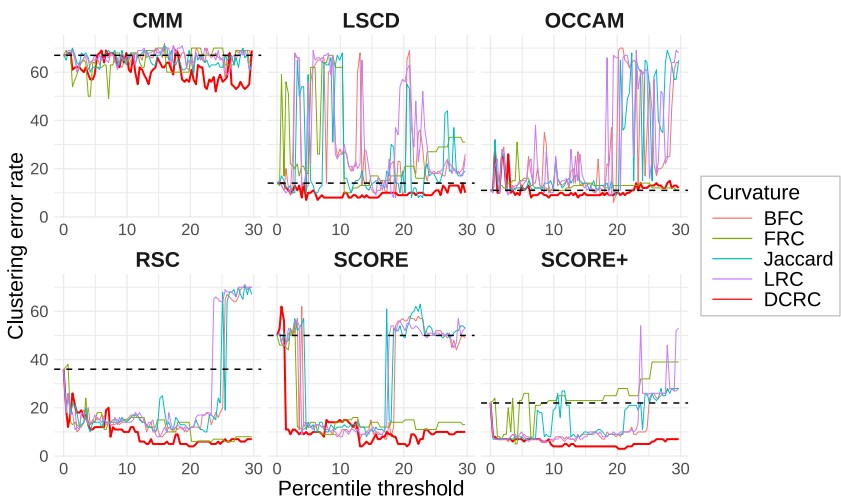

Figure S1: Comparison of curvature performance at various quantile thresholds for six community detection algorithms.

Figure S2 shows the distribution of BFC, FRC, LRC, and DCRC for a single instance of each of the seven graphs listed in Table S1. In each histogram, red bars represent across-community edges, while and blue bars indicate within-community edges. In Graph 1, 3, and 4, DCRC demonstrates clearer separation between across-community edges and within-community edges compared to the other curvature metrics. Notably, FRC values exhibit more overlap between edge types than others. In the remaining histograms, although there is a substantial overlap between across- and within-community edges - highlighting the difficulty of community detection in these graphs - the DCRC-based preprocessing method still leads to improved community detection performance.

Table S1: Parameters used to generate seven DCBM graphs in Experiment 3.

| Dataset | Parameters (x, y, A, B) |
|---------|-------------------------|
| Graph 1 | (0.5, 1.5, 0.6, 0.3) |
| Graph 2 | (0.5, 1.5, 0.18, 0.06) |
| Graph 3 | (0.5, 1.5, 0.9, 0.3) |
| Graph 4 | (0.5, 2.5, 0.6, 0.3) |
| Graph 5 | (0.1, 0.8, 0.9, 0.3) |
| Graph 6 | (0.5, 2.5, 0.6, 0.3) |
| Graph 7 | (0.1, 0.8, 0.6, 0.3) |

| Metrics | $n$ | Mean | SD | QQ Corr |
|---------|-----|------|-----|---------|
| | 1000 | -0.072 | 0.997 | 0.998 |
| $T_{4,n}$ | 3000 | -0.039 | 1.000 | 0.999 |
| | 5000 | -0.029 | 1.003 | 1.000 |
| | 1000 | -0.259 | 1.096 | 0.968 |
| $T_{5,n}$ | 3000 | -0.131 | 0.957 | 0.993 |
| | 5000 | -0.099 | 0.940 | 0.997 |

Table S2: Summary statistics from the simulations for Theorem 4 and Theorem 5. For a DCBM network with $n$ total nodes, the sample mean, standard deviation, and QQ plot correlation are reported in the table.

**Experiment 4 - Simulation for Theorem 4 and Theorem 5:** For empirical validation of Theorem 4 and Theorem 5, we generated three DCBM networks with increasing total node counts ($n = 1000, 3000, 5000$), holding $(K, x, y, A, B) = (2, 1, 1, 0.07, 0.1)$ and $\gamma = (0.5, 0.5)$. As shown in Figure S3, the distribution of $T_{4,n}$ and $T_{5,n}$ converges to standard normal distribution as the total number of counts increases.

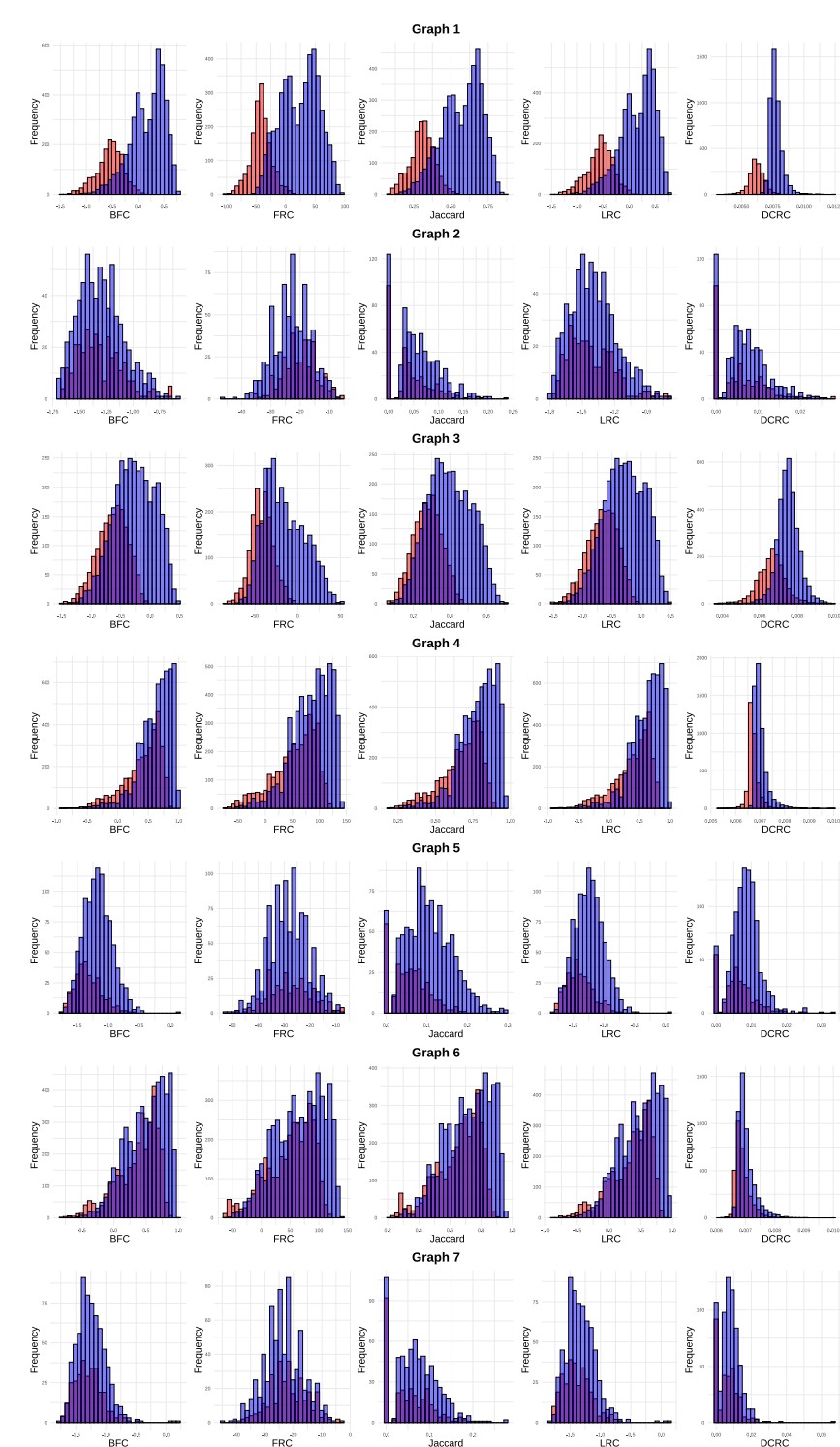

Figure S2: Curvature histograms for seven graphs used in Experiment 3.

Table S2 presents summary statistics, where we can observe that the sample mean of normalized DCRC values approaches zero and its standard deviation converges to one as $n$ increases. Moreover, the correlation of QQ plot quantiles approaches one as the total number of nodes increases. Hence,

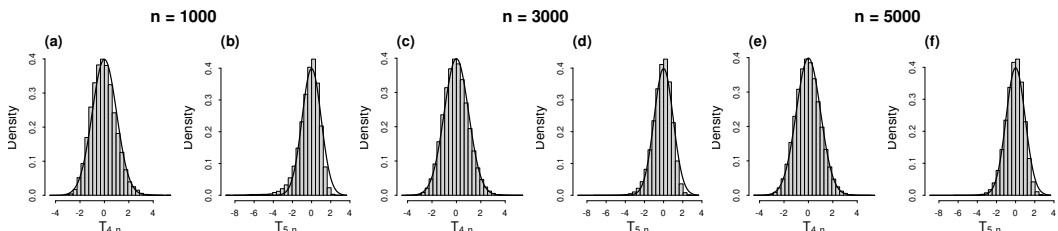

Figure S3: Simulated distributions of the metrics from Theorem 4 ($T_{4,n}$) and Theorem 5 ($T_{5,n}$) across increasing network sizes $n$. The overlaid solid line is the standard normal distribution, $N(0,1)$.

these results empirically validate the asymptotic normality property of DCRC in Theorem 4 and Theorem 5.

**Experiment 5 - Additional Results for Varying Total Node Number:** We evaluated our method on DCBM networks with increasing numbers of nodes, each with eight communities in size ratios $(0.04, 0.07, 0.11, 0.14, 0.18, 0.21, 0.25, 0.29)$. Following the notation of Table S1, we set the heterogeneity and probability matrix parameters to $(x, y, A, B) = (0.5, 1.5, 0.6, 0.3)$ and follow the pipeline in Experiment 1. To further show the robustness, we generated a hundred independent graphs per configuration, and presented the improvement of clustering error rate due to DCRC-based preprocessing relative to the baseline in Figure S4. Each improvement was normalized by its total node number, for fair comparison between different graph sizes. The results show that DCRC significantly improves all six methods across all three node sizes.

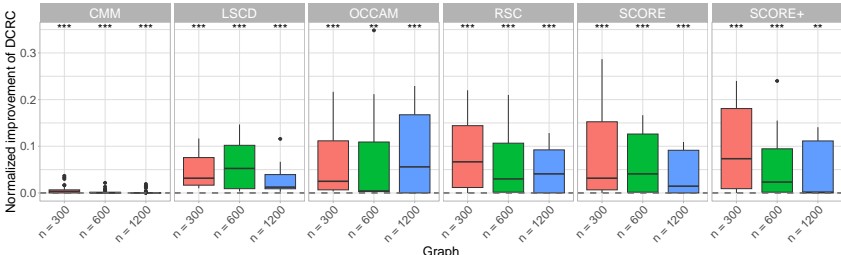

Figure S4: Box plots illustrating the performance improvement, quantified as the reduction in the clustering error rate, from DCRC-based preprocessing over the baseline as total number of nodes changes.

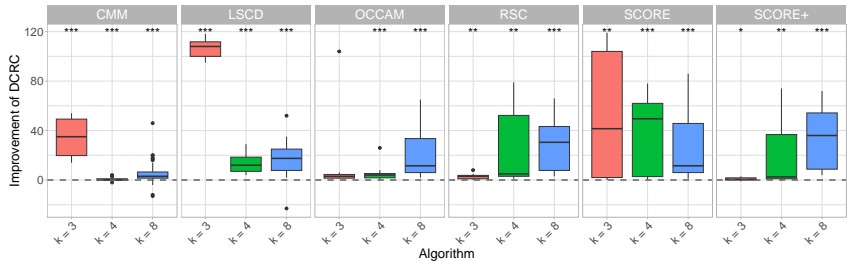

Figure S5: Box plots summarizing the performance improvement, quantified as the reduction in the clustering error rate, from DCRC-based preprocessing over the baseline as total community number changes.

**Experiment 6 - Additional Results for Varying Community Number:** We assessed our method on DCBM networks with increasing numbers of community, keeping $n = 300$ fixed. Following the notation of Supplement Table S1, we set $(x, y, A, B) = (0.5, 1.5, 0.6, 0.3)$ and carried out the experiments as in Experiment 1. Specifically, we tried $K = 3, 4, 8$, to match the number of communities

in real dataset used in our manuscript, with community ratio $(0.1, 0.2, 0.7)$, $(0.10, 0.20, 0.30, 0.40)$, and $(0.04, 0.07, 0.11, 0.14, 0.18, 0.21, 0.25, 0.29)$. For each configuration, we generated a hundred independent graphs and report the distribution of improvement of clustering error relative to the baseline in Figure S5. DCBM significantly improves all community detection methods by reducing mismatch counts regardless of the number of communities, except for OCCAM with $k = 3$. However, the extend of improvement varies across methods and number of communities, likely due to differences in the nature of the existing community detection methods.

### D.2  REAL DATA

Figures S6, S7, and S8 illustrate the change in clustering error rate from the baseline across different threshold line. Since real data does not exactly follow the DCBM model, we also utilized the variability of DCRC values in the threshold. This threshold enables the removal of edges with both low DCRC values and low variability, increasing the likelihood of eliminating consistently low DCRC edges. The detailed procedure is as follows: each threshold line is drawn on a scatter plot where x-axis represents DCRC value and y-axis represents the standard deviation of $\mathrm{DCRC}(i, j)$. Edges lying below the threshold line are removed from the graph during preprocessing. The *Theta* in the y-axis of those figures indicates the angle between the threshold line and the x-axis, while the *Intercept* represents its x-intercept. In the heatmaps, red regions indicate improvement over the baseline, while blue regions indicate a decline in performance. From each heatmap, we observe that after a certain combination of slope and x-intercept, the community detection results either improve or remain comparable to the baseline. The values reported in Table 1 of the main text highlight the best improvements achieved through DCRC-based preprocessing for each community detection algorithm.

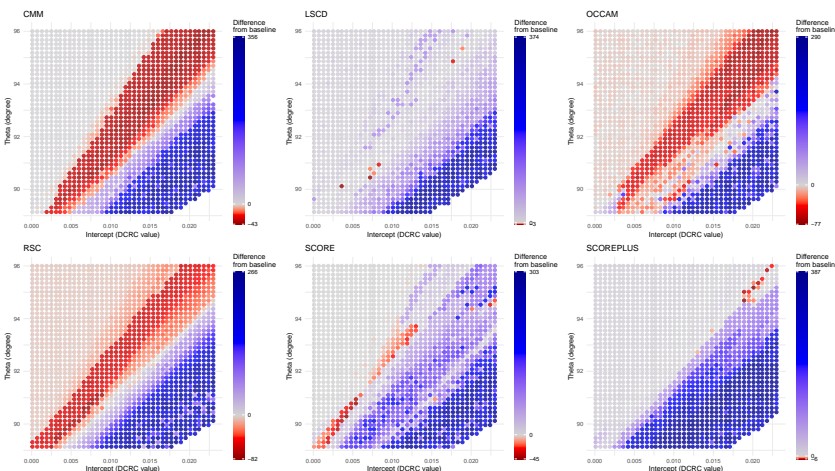

Figure S6: Heatmap of clustering error rate differences from baseline on the Caltech dataset, across combinations of angle of the threshold line and x-intercept.

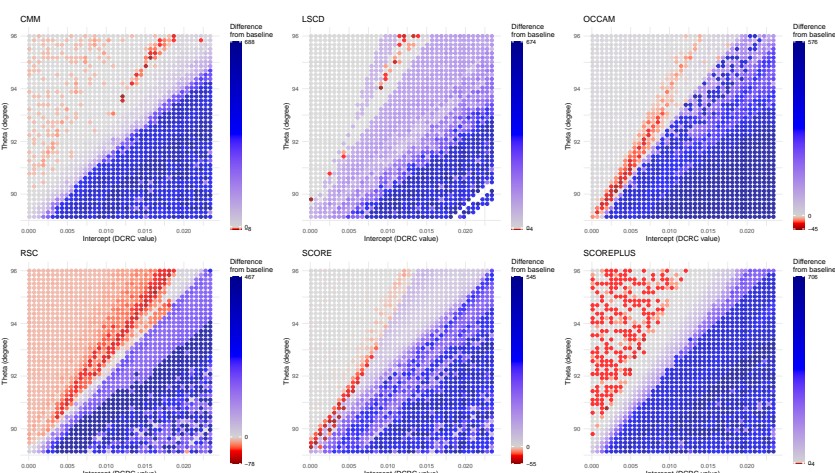

Figure S7: Heatmap of clustering error rate differences from baseline on the Simmons dataset, across combinations of angle of the threshold line x-intercept.

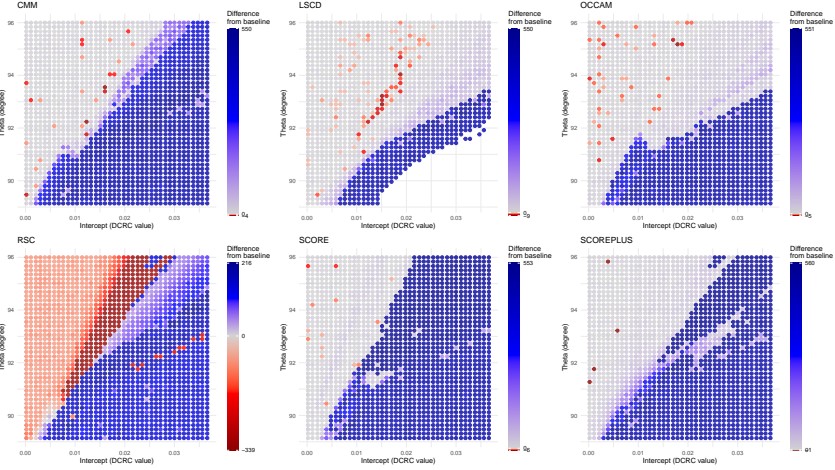

Figure S8: Heatmap of clustering error rate differences from baseline on the Political blog dataset, across combinations of angle of the threshold line x-intercept.

