# OpenReview forum: "Degree-Corrected Ricci Curvature for Networks"
_ICLR.cc/2026/Conference — ICLR 2026 Conference Withdrawn Submission_

### Official Review · Reviewer_uZBP · 2025-10-29

**Soundness:** 2
**Presentation:** 1
**Contribution:** 2
**Rating:** 2
**Confidence:** 3

**Summary:**

This paper presents a new mode of discrete curvature and applies it in a community detection problem in the context of the stochastic block model and degree corrected block model.

**Strengths:**

- There is quite a wide range of curvature modes tested, though I would have liked to see more (see Questions below).
- Good range of community detection methods tested after preprocessing.

**Weaknesses:**

- I am not sure about the claim of the authors that their work is the first to study curvature in the context of random graphs, see "Comparative analysis of two discretizations of Ricci curvature for complex networks," Samal et al., Scientific Reports volume 8, Article number: 8650 (2018).  Indeed, the questions and characteristics studied are different, but I think that it is not correct to claim to be the first such study and position it as a main theoretical contribution.
- The paper is not well written and there were many grammatical errors throughout (e.g., "optimal transpose" instead of "transport").
- There appears to be no complexity or convergence analysis.
- There is not much variation in sizes of networks and so difficult to assess the performance on varying graph sizes and characteristics (such as topological complexity).

**Questions:**

- Have you considered the augmented Forman and Haantjes curvatures?
- There is a brief discussion on the computational runtime but there is a vast difference in computing the different modes of curvature studied in the work and it is not clear how each curvature type tested affects the final runtime.  In particular, it is not clear what the computational cost of computing the newly proposed curvature mode is.

---

### Official Review · Reviewer_ezgp · 2025-10-31

**Soundness:** 2
**Presentation:** 2
**Contribution:** 1
**Rating:** 2
**Confidence:** 3

**Summary:**

This paper addresses the limitations of existing network curvature measures when applied to networks with degree heterogeneity. The authors show that existing curvature measures satisfy the "diagonal-dominating property" under the Stochastic Block Model (SBM) but fail under the Degree-Corrected Block Model (DCBM). To address this, they propose Degree-Corrected Ricci Curvature (DCRC), defined as the normalized ratio of triangles attached to an edge. The paper provides theoretical analysis showing DCRC maintains the diagonal-dominating property under both SBM and DCBM, analyzes the properties of DCRC, and demonstrates through simulations and real-world experiments that DCRC-based preprocessing can improve existing community detection algorithms in specific settings.

**Strengths:**

**S1.** The paper identifies a limitation of existing curvature measures under degree heterogeneity.

**S2.** The theoretical analysis is substantial, providing population-level analysis and asymptotic distributions with uncertainty quantification.

**Weaknesses:**

**W1.** To my understanding the major weakness is lack of practical motivation. The paper addresses a very specific problem: satisfying the diagonal-dominating property for curvature-based community detection specifically under DCBM. The authors fail to establish why this narrow focus should be of interest to the community at a machine learning venue like ICLR. Not only the ML contribution is unclear, the practical relevance of this in network analysis in general is unclear (I do not find the experiments on real-world networks convincing to show the practical relevance).

**W2.** The proposed DCRC seems to be an edge-based analog of the well-established local clustering coefficient for the nodes. An unnormalized version of this, referred to as “edge multiplicity” has been in fact studied back in 2006 [1]. The authors don't acknowledge this connection. Moreover, the paper doesn’t clarify why this is referred to as a type of “Ricci curvature”. Ricci curvature characterizes deviation of local geometry from being Euclidean, and it measures volume change as a volume ball travels along a geodesic. The paper should explain how DCRC captures this in graphs.

**W3.** The theoretical results rely on assumptions that make the work less practical in common empirical settings. In particular, on line 251 the authors state that their assumption requires that "the degree parameter of any single node cannot be excessively dominant with respect to
other nodes". My understanding is that this assumption is violated with a power-law degree distribution which is prevalent in real-world networks.

**W4.** The experiments are predominantly on a limited class of synthetic networks. The real-world network experiments are limited and use a preprocessing step (removing edges with low DCRC and low variance) that is not properly explained or justified.


**W5.** A minor point: The authors claim that ORC is “less amenable to theoretical analysis”. This is not true to my understanding; especially considering the optimal transport formulation that relates it to well-studied properties of the Wasserstein space of probability measures.

[1] Serrano, M. Á., & Boguna, M. (2006). Clustering in complex networks. I. General formalism. Physical Review E—Statistical, Nonlinear, and Soft Matter Physics, 74(5), 056114.

**Questions:**

**Q1.** How does DCRC relate to the clustering coefficient? Specifically, can you clarify how/whether you can derive one from the other?

**Q2.** Related to W2, can you clarify why this is a notion of Ricci curvature?

**Q3.** Can you elaborate on how and why you preprocess the network in your experiments on real-world networks and remove the low-curvature edges?

---

### Official Review · Reviewer_UKNG · 2025-10-31

**Soundness:** 4
**Presentation:** 4
**Contribution:** 4
**Rating:** 8
**Confidence:** 5

**Summary:**

This paper introduces Degree-Corrected Ricci Curvature (DCRC), a novel graph curvature metric designed to remain informative in the presence of degree heterogeneity. The work formalizes curvature behavior under both SBM and DCBM, proving that DCRC uniquely satisfies a diagonal-dominating property even when node degrees vary substantially. The authors derive population-level expressions, large-deviation bounds, and asymptotic normality results enabling uncertainty quantification. DCRC is then used as a preprocessing mechanism to improve downstream community detection. Extensive simulations and experiments on real-world networks (Caltech, Simmons, Political Blogs) demonstrate consistent improvement of six community detection algorithms over baselines and existing curvature measures.

**Strengths:**

1. The theoretical development is rigorous and clearly motivated. The proof strategy is detailed and leverages established DCBM theory. Large-deviation bounds and asymptotic normality significantly strengthen the contribution. Empirical evaluations are thorough in both synthetic and real benchmarks.
2. The paper is clearly written, well-structured, and provides useful intuition alongside theory. Figures and tables effectively communicate results. Some mathematical exposition is dense and may be challenging for readers without prior exposure to random graph asymptotics, but overall clarity is strong.
3. This work addresses a fundamental and previously under-studied gap: curvature-based methods degrade under degree heterogeneity, which is ubiquitous in real networks. DCRC makes curvature tools viable for realistic graph regimes and provides new theory for curvature in random graph models. The combination of theoretical guarantees, statistical inference tools, and applied gains is compelling.

**Weaknesses:**

1. The definition of DCRC is elegant, but the intuition could be clarified earlier. The argument about cancellation of degree effects is correct, yet could be enhanced with geometric intuition, connections to Wasserstein curvature, or brief toy examples illustrating why traditional curvature fails.

**Questions:**

1. How sensitive is DCRC to extremely sparse regimes (near connectivity threshold)?
2. Can the authors comment on the runtime cost relative to FRC/Jaccard beyond ORC?
3. Can you discuss other applications of curvature except for community detection?

Clarifications would enrich the impact and applicability discussion.

---

### Official Review · Reviewer_rHh9 · 2025-11-01

**Soundness:** 1
**Presentation:** 3
**Contribution:** 1
**Rating:** 0
**Confidence:** 5

**Summary:**

The paper proposes Degree-Corrected Ricci Curvature (DCRC) for edges, defined as n_ij/(n_in_j), to correct degree-heterogeneity effects when using curvature-like scores for community detection.
The authors prove that the population-level DCRC separates within- vs across-community edges under both SBM and DCBM, give concentration and asymptotic-normality results, and show that pruning edges by low DCRC improves several community-detection methods on simulated DCBM data and three real networks.
Empirical results report some improvements.

**Strengths:**

* DCRC intuitively makes sense as a score, but not as a curvature.
* Paper contains analysis of the proposed score under degree-corrected SBM model assumption.
* Experiments demonstrate DCRC-based edge-pruning improves performance of six diverse community-detection algorithms across many simulated DCBM settings and three real datasets.
* Definitions, assumptions, and paper are largely easy to follow.

**Weaknesses:**

tl;dr: The contributions are overstated, the score is seemingly not a curvature, comparable similarity scores have not been discussed, results are minor when assuming degree corrected SBMs.

* Fundamentally, the paper mischaracterizes the novelty and prior work. The proposed DCRC (n_ij/(n_in_j)) seems to be essentially the Leicht–Holme–Newman (LHN) common-neighbor normalization used for link-prediction. The manuscript does not acknowledge or compare to LHN (or other degree-normalized CN measures), so the claimed novelty as a "Ricci curvature" is misleading.
* Calling a designed-to-be nonnegative similarity score a "Ricci curvature" is conceptually problematic, wrong, and potentially *very* confusing. Established discrete Ricci curvatures (Ollivier, Forman, etc.) can take negative values (to e.g., signify node dispersion/tree-like graphs over a positive curvature of clique-like graphs); DCRC is apparently nonnegative and lacks sign information entirely. The geometric interpretation and terminology are therefore largely unsupported. In sum, dispersion vs concentration is seemingly not properly encoded. Rather it uses a degree-corrected link prediction score.
* No direct empirical comparison to other simple degree-corrected baselines (LHN) or to other link-prediction scores is provided (e.g., Salton Cosine Index); without this the performance gains may be unsurprising and attributable simply to known scores.
* The theoretical results require relatively dense regimes where nij is meaningful; this reduces the potential applicability to many real-world sparse networks. The paper remarks on this, but more empirical characterization of boundaries would help but are not included.
* The theoretical results are seemingly straightforward algebraic derivations assuming a degree corrected SBMs.
* The paper's population separation, concentration bounds, and asymptotic normality for DCRC under a DCSBM follow by straightforward algebraic evaluation of expectations for nij and then application of standard concentration inequalities (Bernstein/Chernoff) and a Lindeberg–Feller CLT; therefore the contribution is mainly a careful, useful instantiation of well-known probabilistic tools to a simple degree-normalized common-neighbour statistic.
* Labels in at least Fig. 1--5 are minuscule and are hard to read.
* Motivation behind experimental setup not very clear.

**Questions:**

- Do not oversell the score as Ricci curvature.
- Contextualize with established measures (e.g., LHS)
- Improve plots (re label size).

---

### Note · Authors · 2025-11-21

I have read and agree with the venue's withdrawal policy on behalf of myself and my co-authors.